# Evidence for an increase in cannabis use in Iran – A systematic review and trend analysis

Yasna Rostam-Abadi[1], Jaleh Gholami[1], Masoumeh Amin-Esmaeili[1,2],
Shahab Baheshmat[1,3], Marziyeh Hamzehzadeh[1,3], Hossein Rafiemanesh[4],
Morteza Nasserbakht[5], Leila Ghalichi[6], Anousheh Safarcherati[1], Farhad Taremian[7],
Ramin Mojtabai[2], Afarin Rahimi-Movaghar[1] *

1 Iranian National Center for Addiction Studies (INCAS), Tehran University of Medical Sciences, Tehran,
Iran, 2 Mental Health Department, Johns Hopkins Bloomberg School of Public Health, Baltimore, Maryland,
United States of America, 3 Department of Neuroscience and Addiction Studies, School of Advanced
Technologies in Medicine (SATiM), Tehran University of Medical Sciences, Tehran, Iran, 4 Student Research
Committee, Department of Epidemiology, School of Public Health and Safety, Shahid Beheshti University of
Medical Sciences, Tehran, Iran, 5 Mental Health Research Center, Tehran Psychiatry Institute, Iran
University of Medical Sciences, Tehran, Iran, 6 Mental Health Research Center, Psychosocial Health
Research Institute, Iran University of Medical Science, Tehran, Iran, 7 Substance Abuse and Dependence
research Center, University of Social Welfare and Rehabilitation Sciences, Tehran, Iran

* rahimia@tums.ac.ir

pone.0256563

University Faculty of Medicine, THAILAND

**Data Availability Statement:** All relevant data are
within the manuscript and its Supporting
Information files.

## Abstract

### Background and aims

Cannabis is the most widely used illicit substance globally. In this systematic review,
we examined the prevalence and trends of cannabis use and cannabis use disorder in
Iran.

### Methods

We searched International and Iranian databases up to March 2021. Pooled preva-
lence of use among sex subgroups of the general population, university and high
school students, combined youth groups, and high-risk groups was estimated through
random-effects model. Trends of various use indicators and national seizures were
examined.

### Results

Ninety studies were included. The prevalence estimates of last 12-month cannabis use
were 1.3% (95%CI: 0.1–3.6) and 0.2% (95%CI: 0.1–0.3) among the male and female Ira-
nian general population, respectively. The prevalence of cannabis use disorder among
general population in national studies rose from 0% in 2001 to 0.5% in 2011. In the 2016–
2020 period, the pooled prevalence estimates of last 12-month cannabis use were 4.9%
(95% CI: 3.4–6.7) and 0.3% (95% CI: 0.0–1.3) among males and females of "combined
youth groups", respectively. The linear trend of last 12-month cannabis use among males
of "combined youth groups" and among female university students increased significantly
from 2000 to 2020.

**Funding:** This study was supported financially by the Iranian National Institute for Medical Research Development (NIMAD), (Grant No. 940043 to ARM). The funding source had no role in the study design, data synthesis, interpretation of the data, and in the drafting of the manuscript. http://nimad. ac.ir/.

**Competing interests:** The authors have declared that no competing interests exist.

## Conclusions

Prevalence of cannabis use in Iran is low compared to many countries. However, there is strong evidence of an increase in cannabis use among the youth and some evidence for an increase in cannabis use disorder.

## Introduction

Cannabis is the most widely used and trafficked illicit substance in the world with 192 million cannabis users globally in 2018 [1]. The prevalence of cannabis use in the last 12-month has been increasing in the last decade, reaching 3.9% of individuals aged 15–64 years [1]. Cannabis use has been legalized and regulated in several countries in recent years and the effect of policy changes on the extent of use and its health consequences are under close monitoring.

Cannabis abuse and dependence potential have been demonstrated, mainly linked to tetra-hydrocannabinol (Δ9-THC) concentrations, the main psychoactive constituent of cannabis [2]. Some estimates indicate that one-tenth of cannabis users can become dependent [3, 4]. Moreover, adverse effects on brain development, acting as a gateway drug, and triggering psychiatric disorders have been linked to the regular and early age of cannabis use [5]. Low birthweight, motor vehicle injuries, and bronchitis are also among the health-related harms associated with recreational cannabis use [6].

While opium is the main illicit drug used in Iran, cannabis has also been used for a long time in the country. The use of cannabis goes back at least to the 16th century when cannabis was used in types of religious ceremonies by Sufis [7]. Currently, there is no licit or medical production of cannabis in Iran, and the rulings have considered a strict prohibition on its use [8]. However, there are some concerns that cannabis use is increasing in the country and is becoming an important public health problem. Several studies have examined the prevalence of cannabis use along with other drugs in the general population. Previously, a systematic review was conducted up to 2014 on the lifetime cannabis use among Iranian university and high-school students [9]. However, we know little about the prevalence of cannabis use in other Iranian population subgroups, other use indicators among various populations, and the extent of cannabis use disorder in Iran. This study aimed to use the available data to provide 1) the estimate of cannabis use (lifetime, last 12-months, last month and current, daily or almost daily use), 2) the estimate of cannabis use disorder, both in the subgroups of Iranian population (general population, youth, university students, high school students, and high-risk groups), and 3) the trends of estimates until 2020.

## Methods

### Search strategy

Three international databases (Web of Science, Scopus, and PubMed) and an Iranian database, the Scientific Information Database (SID), were searched from January 1990 up to 16 March 2021. As the first legislation changes on cannabis use in the countries initiated in 1990s, we extended our search limit to 1990 to be able to investigate the trend. Furthermore, we 1) hand-searched the reference list of the retrieved scientific documents (backward citation tracking), 2) communicated with experts in the field of addiction (principle investigators of national or large studies) in Iran to access unpublished studies, such as thesis and unpublished reports, 3) hand-searched final reports of studies of Drug Control Headquarters' resources, and 4) hand-

searched the Iranian National Center of Addiction Studies (INCAS) archives on Iranian epide-miological studies.

Search strategy (S1 Table) for the international databases was developed using three groups of key-terms which were combined using Boolean operators: 1) general terms related to drug use or drug use disorder; 2) the names of substances commonly used in Iran including differ-ent forms of cannabis, opioids, stimulants, and alcohol; 3) keywords related to Iran, including names of provinces and major cities. Keywords related to other substances were added to the search strategy in order not to miss relevant studies without cannabis-related terms in the title or abstract. No restrictions were applied to the study design. The Iranian database was searched only with the Persian and English words for different forms of cannabis.

### Eligibility criteria and screening

All studies providing the prevalence of cannabis use or use disorder among the Iranian popula-tion were included. Whatever criteria of cannabis use disorder were applied, the studies were included -either based on Diagnostic and Statistical Manual of Mental Disorders version IV or V or any other definitions. The applied criteria were reported exactly as stated in the study.

The eligible target population was the general population, university students, high school students, and the high-risk population. Based on our previous reviews [10, 11], these groups were the main targets investigated in prevalence studies and therefore were selected. Any pop-ulation representative of the Iranian population and not considered high-risk for substance use and use disorder was classified as "general population", including population being sam-pled in household surveys, from public places, in industrial settings, or health centers irrele-vant to substance use. Some studies on the general population recruited only youths, and some included a population of 15 years and over or 18 years and over. Therefore, we requested the authors of the latter studies to provide age-group specific data and we created a separate popu-lation category, "young general population", with a wide age definition of 15–34 years. Any specific population that was assumed to have with higher rates of substance use and use disor-der than the general population was categorized as a "high-risk population".

Studies were excluded if the use or use disorder indicator was not reported or unclear, the prevalence of different types of cannabis (resin and plant) was reported separately without reporting the merged prevalence of any cannabis use or use disorder, if was case-control or interventional study, and the source population was not eligible.

Screening of the retrieved documents was carried out in two stages: screening of the titles and abstracts for including all relevant studies and assessment of the full texts for eligibility cri-teria. Two different reviewers (MAE, SB, MH, YRA, and HR) conducted both stages indepen-dently, and inconsistencies were resolved by a third reviewer (ARM).

### Data extraction and quality assessment

For each included study, the following data were extracted: first author, publication year, the language of the manuscript, the year of the study implementation, recruitment setting(s), tar-get population, study location (province), sampling method, sample size, response rate, age characteristics of the participants, use indicator(s), criteria used for diagnosis of use disorder, and finally the prevalence of cannabis use and use disorder in each sex subgroup. Quality of the included studies was assessed using a 9-item rating adapted from Joanna Briggs Institute quality assessment tool [12] and previously used in other studies by our group [10, 11] (S2 Table). Two different investigators (SB, MH, YRA, and HR) independently extracted data from the included studies, and the discrepancies were resolved through discussion with a third reviewer (ARM).

## Statistical analysis

Characteristics of all included studies, their findings on the prevalence of cannabis use and use disorder, and the results of quality assessment of each included study were recorded in tables separately for the general population, university students, high school students, and high-risk populations (including people who use drugs (PWUD), prisoners, and other high-risk groups).

All eligible studies, which reported prevalence separately in the two sexes, were included in the meta-analysis. Studies not reporting sex-specific data were not included in the meta-analysis. The overall prevalence of cannabis use was estimated using the "metaprop" command ("metafor" package) separately by sex, population subgroups (general population, young general population, university students, high school students, and high-risk groups), timeframe and frequency of use (lifetime, last 12-month, last month or current, daily or almost daily, current main drug), and study year (2000–2005, 2006–2010, 2011–2015, and 2016–2020). The studies conducted before 2000 did not provide sex-specific data therefore were not entered in the analyses. The pooled prevalence estimates in each sex and population subgroups were presented using separate forest plots. Random-effects models were used for pooling the estimates and Freeman–Tukey double arcsine transformation was used for stabilizing the variance. The heterogeneity between studies was quantified by the $I^2$ statistic. We also conducted meta-regression analyses via the "metareg" command ("metafor" package) to examine the association between the prevalence of cannabis use and several covariates including sex, timeframe and frequency, study year, number of unmet quality criteria, and study population (young general population, university students, high school, and high-risk population, all versus the general population). We broke down studies providing estimates among both sexes or on various timeframes and frequencies and regarded them as separate studies in the model. If studies were based on network scale-up (NSU) method (an indirect estimation by measuring the respondents' networks size and the number of cannabis users in their network [13]), we excluded them from the meta-analysis model and only presented them in the relative tables so that the results would be comparable. Moreover, to assess the effect of quality of studies on pooled estimates, sensitivity analyses were performed by removing studies with more than two unmet items on the quality scale.

Due to the scarcity of data for some periods, we merged studies among the young general population, university students, and high school students under the "combined youth groups" category for trend plot. We categorized studies into four periods as follows: 2000–2005; 2006–2010; 2011–2015; and 2016–2020; in order to have enough data points for trend analysis. We plotted the pooled prevalence of the last 12-month, last month or current, and daily or almost daily use of cannabis among males and females of "combined youth groups" for each period using the "ggplot" command ("ggplot2" package). As the heterogeneity among the "combined youth groups" was high and might have obscured trend patterns, we also analyzed the trends in the prevalence of the last 12-month use of cannabis among male and female university students, which had enough numbers in each period using similar methodology. Similarly, among the regional subgroups of the "combined youth groups", there were adequate number of studies only for Tehran province to perform trend analysis. We were not able to provide a trend plot for studies conducted among the general population due to the limited number of studies in each period. The pooled estimates are presented in the middle of each period. We fitted meta-regression lines for assessing the significance of the slope of the trend lines. Moreover, the data on national seizures of cannabis in 100 metric tonnes (annually from 1990 to 2018) [14] are presented in the trend plot for better interpretation of the results.

All statistical analyses were performed using R statistical software (version 4.0.3) and geographical distribution map of the prevalence of the last 12-month cannabis use among

"combined youth groups" was provided using ArcGIS software (version 10.5). The Preferred Reporting Items for Systematic Reviews and Meta-Analyses (PRISMA) statement was used for reporting this systematic review and meta-analysis study.

## Results

Through the search of international databases, and after excluding the duplicates, titles and abstracts of 3,686 records were reviewed (Fig 1). Of all these records, 285 were eligible for full-text review. Additionally, from 2,530 records found in the SID, only four records were eligible for full-text review. Through contact with experts, backward citation tracking and other opportunistic methods, 50 other studies were also identified. A total of 90 studies were included in the final sample providing the prevalence of cannabis use or use disorder among the general population (N = 12), young general population (N = 9), university students (N = 33), high school students (N = 18), and high-risk groups (N = 22). Four studies provided measures for both the general population and young general population. Overall, 37.8% of the reports were in Persian and the remaining 62.2% were in English. Among the 50 studies included through opportunistic methods, 17 were not published in peer-reviewed journals (two unpublished studies, six theses, and nine final reports of studies). From these 17 studies, only one study had more than two unmet quality criteria that was not included in the meta-analysis as sex-specific data was not reported. In total, one study was excluded from the meta-analysis due to the application of the NSU method, and nine studies as sex-specific data were not reported. The characteristics and results of the studies are presented in Tables 1–4 based on the target population.

### General population

Twelve studies provided the prevalence of cannabis use among the Iranian general population with a total sample size of 131,345 (53.7% male) from 2001 to 2016 (Table 1). Six studies were conducted nationally between 2001 and 2015 and the other six studies were conducted in three different provinces. Eight reports were based on household surveys, and the others recruited their samples from the street, industrial plants, hospitals, or a clinical laboratory. The mean age of the participants ranged from 30.8 to 40.2 years. The pooled prevalence estimates of lifetime cannabis use were 2.7% (95%CI: 0.6–6.1; $I^2$ = 98.0%; 5 studies; Table 5 and S1 Fig)

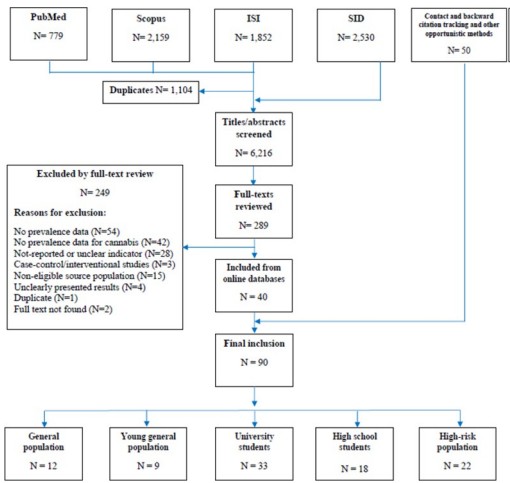

**Fig 1. Flow diagram of study selection.**

**Table 1. Characteristics and results of studies on the prevalence of cannabis use and use disorder among the general population.**

| | Author, Date | Lang | Year of study | Province | Setting/ Participants | Response rate(%) | Age characteristics (year) | Sample size (Male; Female) | Time indicator | Prevalence of use (%) | | | Numerals of unfulfilled quality items[a] |
|---|---|---|---|---|---|---|---|---|---|---|---|---|---|
| | | | | | | | | | | Male | Female | Total | |
| **All ages** | | | | | | | | | | | | | |
| 1 | Najafipour, 2017 [15][b] | En | 2016 | Kerman | Household; individuals aged 15–75 years | NR | Range: 15–75 | 6016 (1956; 4060) | Lifetime | 0.6 | 0 | - | 6 |
| 2 | Damari, 2020 [16] | En | 2015 | National | Employees of industrial plants | 97.3 | Mostly 21–40 years | 13128 (12077; 1051) | Current (self-report) | - | - | 0.2 | 9 |
| | | | | | | | | | Current (Urine test) | - | - | 5.1 | |
| 3 | Noorbala, 2020 [17] | En | 2015 | National | Household; individuals above 15 years | 75.6 | Range: above 15 | 27663 (13796; 13867) | Lifetime | 0.7 | 0.2 | 0.4 | - |
| | | | | | | | | | Last 12-month | 0.6 | 0.2 | 0.4 | |
| 4 | Roshanpajouh, 2020 [18] | En | 2015 | National | Household; individuals aged 15–64 years | 95.5 | Mean 37.1 | 57450 (29185; 28265) | Last week | - | - | 0.4 | 9 |
| 5 | Nikfarjam, 2016 [19] | En | 2013 | National | Street-based; individuals over 18 years; indirect method | NR | Mean: 30.8 Range: 18–87 [c] | 7535 (3584; 3853) | Last 12-month (NSU) | 0.9 | 0.07 | 0.5 | 2, 6, 9 |
| 6 | Ziaaddini, 2013 [20][b] | En | 2012 | Kerman | Household; adult residents in a rural area | 75.0 | Mostly: below 30 | 900 (490; 410) | Lifetime | 2.7 | 1.2 | 2.0 | 2, 9 |
| | | | | | | | | | Last month | 1.0 | 0.7 | 0.9 | |
| | | | | | | | | | Daily or almost daily | 0.8 | 0.2 | 0.6 | |
| 7 | Amin-Esmaeili, 2016 [21] | En | 2011 | National | Household; individuals aged 15–64 years; self-administered questionnaire | 85.7 | Range: 15–64 | 3437 (1514; 1923) | Last 12-month | 2.4 | 0.2 | 1.3 | - |
| | | | | | Household; individuals aged 15–64 years; Interview | 85.7 | Range: 15–64 | 7841 (3366; 4475) | Five times and more in the last 12 months | 1.7 | 0.1 | 0.9 | |
| | | | | | Household; individuals aged 15–64 years; self-administered questionnaire | 85.7 | Range: 15–64 | 3437 (1514; 1923) | Daily or almost daily | 0.9 | 0 | 0.4 | |
| | | | | | Household; individuals aged 15–64 years; Interview | 85.7 | Range: 15–64 | 7841 (3366; 4475) | Use disorder in the last 12-month diagnosed based on DSM-IV criteria | 1.0 | 0 | 0.5 | |

(*Continued*)

**Table 1.** (Continued)

| | Author, Date | Lang | Year of study | Province | Setting/ Participants | Response rate(%) | Age characteristics (year) | Sample size (Male; Female) | Time indicator | Prevalence of use (%) | | | Numerals of unfulfilled quality items[a] |
|---|---|---|---|---|---|---|---|---|---|---|---|---|---|
| | | | | | | | | | | Male | Female | Total | |
| 8 | Eftekhar Ardebili, 2006 [22] | En | 2004 | Tehran | Household; individuals over 15 years resided in 6th district | NR | Mean (SD) 40.2 (17.4) | 2685 (1166; 1519) | Last month | 0.3 | 0 | 0.2 | 6 |
| 9 | Rahimi-Movaghar, 2007 [23] | Pe | 2004 | Kerman | Household; Bam earthquake survivors over 14 years | 99.1 | Mean (SD) 31.7 (12.9) | 779 (219; 560) | Lifetime [d] | 2.7 | 0 | - | - |
| | | | | | | | | | Last month [d] | 1.4 | 0 | - | |
| 10 | Ahmadi, 2003 [24][b] | En | 2003 | Fars | Household; individuals over 14 years | 93.3 | Mean (SD) Male 34.6 (14.1); Female 31.0 (12.9) Range: 15–83 | 1400 (700; 700) | Lifetime | 11.4 | 1.1 | 6.3 | - |
| | | | | | | | | | Use disorder in the last 12-month diagnosed based on DSM-IV criteria | 4.3 | 0.7 | 2.5 | |
| 11 | Meimandi, 2005 [25] | En | 2002 | Kerman | Men over 15 years referring to a clinical laboratory | NR | Min: 15 | 694 (694; 0) | Current [e] | 0.6 | - | - | 2, 4, 6 |
| 12 | Yasamy, 2002 [26] | Pe | 2001 | National | Clients of emergency wards; individuals over 14 years | 97.9 | Min: 15 years | 5254 (3341; 1913) | Current | 1.1 | 0.06 | 0.7 | - |
| | | | | | | | | | Current use disorder diagnosed based on DSM-IV criteria | 0 | 0 | 0 | |
| **Ages 34 years and below** | | | | | | | | | | | | | |
| 1 | Rahimi-Movaghar, Unpublished [27] | NA | 2018–20 | Mazandaran | Household; individuals aged 15–34 years; first round | 100 | Mean (SD) 25.6 (6.1) | 2576 (951; 1625) | Lifetime | 7.7 | 0.2 | 4.0 | - |
| | | | | | | | | | Last 12-month | 3.4 | 0.1 | 1.8 | |
| 2 | Rahimi-Movaghar, Unpublished [27] | NA | 2019–20 | Fars | Household; individuals aged 15–34 years; third round | 88.6 | Mean (SD) 31.0 (5.2) | 670 (253; 417) | Daily or almost daily | 1.6 | 0 | - | - |
| 3 | Rahimi-Movaghar, Unpublished [27] | NA | 2019–20 | Kermanshah | Household; individuals aged 15–34 years; third round | 66.9 | Mean (SD) 31.5 (5.1) | 803 (335; 468) | Daily or almost daily | 0 | 0 | - | - |

(*Continued*)

**Table 1.** (Continued)

| | Author, Date | Lang | Year of study | Province | Setting/ Participants | Response rate(%) | Age characteristics (year) | Sample size (Male; Female) | Time indicator | Prevalence of use (%) | | | Numerals of unfulfilled quality items[a] |
|---|---|---|---|---|---|---|---|---|---|---|---|---|---|
| | | | | | | | | | | Male | Female | Total | |
| 4 | Rahimi-Movaghar, Unpublished [27] | NA | 2017–19 | Kerman | Household; individuals aged 15–34 years; first round | 100 | Mean (SD) 25.8 (6.1) | 3006 (1322; 1683) | Lifetime | 12.8 | 0.2 | 6.5 | - |
| | | | | | | | | | Last 12-month | 4.7 | 0.1 | 2.4 | |
| 5 | Rahimi-Movaghar, Unpublished [27] | NA | 2015–17 | Fars | Household; individuals aged 15–34 years; first round | 100 | Mean (SD) 26.3 (5.4) | 3014 (1268; 1746) | Lifetime | 12.8 | 0.2 | 6.5 | - |
| | | | | | | | | | Last 12-month | 7.3 | 0 | 3.7 | |
| 6 | Rahimi-Movaghar, Unpublished [27] | NA | 2015–17 | Kermanshah | Household; individuals aged 15–34 years; first round | 100 | Mean (SD) 27.0 (5.1) | 2991 (1335; 1656) | Lifetime | 4.0 | 0 | 2.0 | - |
| | | | | | | | | | Last 12-month | 1.8 | 0 | 0.9 | |
| 7 | Dolatshahi, 2016 [28] | En | 2014 | Tehran | Street-based; women residing in Tehran between 18 to 25 years | NR | Mean (SD) 21.8 (2.4) | 403 (0; 403) | Last 12-month | - | 3.7 | - | 2, 4, 6 |
| 8 | Ziaaddini, 2013 [20][b] | En | 2012 | Kerman | Household; adult residents in a rural area aged below 30 years | -[f] | Range: 15–34 | 410 (219; 191) | Lifetime | 1.8 | 1.0 | - | 2, 9 |
| | | | | | | | | | Last month | 0.5 | 1.0 | - | |
| | | | | | | | | | Daily or almost daily | 0 | 0.5 | - | |
| 9 | Jalilian, 2014 [29] | En | 2011 | Kermanshah | Street-based; individuals aged 15–19 years | 90.0 | Mean (SD) 16.9 (1.22) Range: 15–19 | 148 (148; 0) | Lifetime | 3.4 | - | - | - |
| 10 | Amin-Esmaeili, 2016 [21] | En | 2011 | National | Household; individuals aged 15–34 years; self-administered questionnaire | -[f] | Range: 15–34 | 2108 (916; 1192) | Last 12-month | 3.0 | 0.2 | 1.6 | - |
| | | | | | Household; individuals aged 15–34 years; interview | -[f] | Range: 15–34 | 4767 (2025; 2742) | Five times and more in the last 12 months | 2.2 | 0.1 | 1.2 | |
| | | | | | Household; individuals aged 15–34 years; self-administered questionnaire | -[f] | Range: 15–34 | 2108 (916; 1192) | Daily or almost daily | 1.4 | 0 | 0.7 | |

(*Continued*)

**Table 1.** (*Continued*)

| | Author, Date | Lang | Year of study | Province | Setting/ Participants | Response rate(%) | Age characteristics (year) | Sample size (Male; Female) | Time indicator | Prevalence of use (%) | | | Numerals of unfulfilled quality items[a] |
|---|---|---|---|---|---|---|---|---|---|---|---|---|---|
| | | | | | | | | | | Male | Female | Total | |
| 11 | Hamdieh, 2008 [30] | Pe | 2005 | Tehran | Public places; individuals aged 15–35 years | NR | Range: 15–35 | 8175 (3731; 4444) | Lifetime | 6.1 | 1.8 | 3.8 | 6 |
| 12 | Barooni, 2008 [31][b] | Pe | 2004 | Tehran | Coffee shops; individuals aged 15–25 years | 95.2 | Range: 15–25 | 1903 (895; 1008) | Lifetime | 24.5 | 6.3 | 14.8 | 2 |
| 13 | Eftekhar Ardebili, 2006 [22] | En | 2004 | Tehran | Household; individuals between 15–30 years resided in 6th district | _[f] | Range: 15–30 | 952 | Last month | - | - | 0.3 | - |
| 14 | Rahimi Movaghar, 2007 [23] | Pe | 2004 | Kerman | Household; Bam earthquake survivors over 14 years | _[f] | Range: 15–34 | 425 (131; 294) | Lifetime [d] | 3.8 | 0 | - | - |
| | | | | | | | | | Last month [d] | 1.5 | 0 | - | |

**a:** 1) The source of sampling was well presented and the sample was representative of the target population. 2) The method of sampling was appropriate (random or census). 3) The sample size was adequate (more than 30). 4) The study subjects and the setting were described in detail. 5) The year of the study was stated. 6) The response rate was provided and it was over 70%. If below 70%, the non-responders were not different from respondents in main demographic characteristics. 7) The condition was measured by valid method. 8) Standard criteria were used for the measurement of the condition. 9) Subgroup analyses for sex, recruitment setting, the definition of use, or time indicator were performed.

**b:** personal communication was made for further data.

**c:** age characteristic provided for the recruited sample.

**d:** before the Bam earthquake.

**e:** positive rapid urine test.

**f:** response rate was provided for the all age sample **Abbreviations**: En: English; Lang: Language; NA: Not applicable; NR: Not reported; Pe: Persian; SD: Standard deviation.

in men and 0.3% (95%CI: 0.0–0.7; $I^2$ = 90.6%; 5 studies; Table 5 and S2 Fig) in women. The prevalence estimates of use in the last 12-month were 1.3% (95%CI: 0.1–3.6; $I^2$ = 97.0%; 2 studies) and 0.2% (95%CI: 0.1–0.3; $I^2$ = 0.0%; 2 studies) in men and women, respectively. The pooled prevalence of last month or current cannabis use were 0.8% (95%CI: 0.4–1.2; $I^2$ = 52.4%; 5 studies) in men and 0.1% (95%CI: 0.0–0.3; $I^2$ = 63.7%; 4 studies) in women. The pooled estimates for daily or almost daily use were 0.9% (95%CI: 0.5–1.4; $I^2$ = 0.0%; 2 studies) in men and 0.03% (95%CI: 0.0–0.5; $I^2$ = 68.2%; 2 studies) in women.

Three studies provided the prevalence of cannabis use disorder among the general population in 2001 and 2011- both nationally- and in 2003 in Fars province. The prevalence of cannabis use disorder in national studies rose from 0% in 2001 to 0.5% in 2011 (Table 1).

## Young general population

We found 9 studies spanning years 2004 to 2020 that reported on the prevalence of cannabis use in the general population aged under 34 years with a total sample size of 28,770 (42.0% male) (Table 1). Of these, one study was conducted nationally and the others were conducted in five different provinces. One study was a prospective biennial cohort study in four different

**Table 2. Characteristics and results of studies on the prevalence of cannabis use among the university students.**

| | Author, Date | Lang | Year of study | Province | Setting/ Participants | Response rate (%) | Age characteristics (year) | Sample size (Male; Female) | Time indicator | Prevalence of use (%) | | | Numerals of unfulfilled quality items[a] |
|---|---|---|---|---|---|---|---|---|---|---|---|---|---|
| | | | | | | | | | | Male | Female | Total | |
| 1 | Delavari, 2018 [32] | NA | 2018 | Tehran | Undergraduates of a large governmental medical university | 90.0 | Mean (SD) 20.7 (1.8) Range: 17–37 | 945 (393; 552) | Lifetime | 11.7 | 3.1 | - | - |
| | | | | | | | | | Last 12-month | 8.7 | 2.2 | - | |
| | | | | | | | | | Last month | 5.1 | 0.9 | - | |
| | | | | | | | | | Daily or almost daily | 0.3 | 0 | - | |
| 2 | Yaghubi, 2018 [33][b] | Pe | 2016–17 | National | Undergraduates of non-medical universities in 30 provinces | 98.1 | Mostly under 20 years | 59213 (27913; 31300) | Lifetime | 10.6 | 3.4 | - | 9 |
| | | | | | | | | | Last 12-month | 6.6 | 1.7 | - | |
| | | | | | | | | | Last month | 4.1 | 1.1 | - | |
| 3 | Halimi, 2020 [34] | En | 2016 | Hamedan | Undergraduates of medical and non-medical universities | 92.2 | Mean (SD) 22.5 (4.2) | 461 (198; 267) | Lifetime (direct) | 10.4 | 2.6 | - | - |
| | | | | | | | | | Lifetime (PRM) | 12.6 | 4.1 | - | |
| | | | | | | | | | Lifetime (NSU) | 14.6 | 1.9 | - | |
| 4 | Zahedi, 2018 [35] | En | 2016 | Kerman | Undergraduates and postgraduates of three universities of a range of majors | 83.6 | Mean (SD) 20.5 (1.5) Range: 18–29 | 1730 (1035; 695) | Last 12-month | 4.3 | 0.3 | - | 2 |
| 5 | Pordanjani, 2018 [36][b] | En | 2015 | Yazd | Undergraduates in a medical governmental university | 100 | Mean (SD) 21.9 (2.2) Range: 18–30 | 250 (120; 130) | Current | 3.3 | 0.8 | - | 6, 9 |
| 6 | Safiri, 2016 [37] | En | 2015 | East Azerbaijan | Undergraduates and postgraduates of a governmental medical university | 97.3 | - | 1730 (705; 1025) | Last 12-month | 2.6 | 0.3 | - | - |
| 7 | Moradmand-Badie, 2020 [38] | En | 2014 | Tehran | Undergraduates from seven universities represented all four quadrants of Tehran | 98.0 | Mean (SD) 22.0 (2.7) | 392 (230; 162) | Lifetime | 16.5 | 3.1 | - | - |
| 8 | Mozafarinia, 2017 [39] | En | 2014 | Tehran | Undergraduates in a medical governmental university | 84.4 | Mean: 22.4 | 422 (189; 233) | Lifetime | - | - | 7.1 | 9 |

*(Continued)*

**Table 2.** (*Continued*)

| | Author, Date | Lang | Year of study | Province | Setting/ Participants | Response rate (%) | Age characteristics (year) | Sample size (Male; Female) | Time indicator | Prevalence of use (%) | | | Numerals of unfulfilled quality items[a] |
|---|---|---|---|---|---|---|---|---|---|---|---|---|---|
| | | | | | | | | | | Male | Female | Total | |
| | | | | | | | | | Last 12-month | - | - | 0.9 | |
| | | | | | | | | | Last month | - | - | 0.9 | |
| | | | | | | | | | Daily | - | - | 0.5 | |
| 9 | Sheikhzadeh, 2014 [40] | En | 2013 | Kerman | Grade 2 and over of a large governmental medical university; indirect method | 84.0 | Mean (SD) 21.9 (2.7) [c] | 420 (157; 263) | Last 12-month (PRM model) | 2.0 | 0.7 | - | - |
| | | | | | | | | | Last 12-month (NSU model) | 0.2 | 0 | - | |
| 10 | Abbasi-Ghahramanloo, 2018 [41] | En | 2012–13 | Tehran | Undergraduates of a large governmental medical university | 89.7 | Mean (SD) 21.1 (3.1) Range: 16–44 | 1985 (609; 1376) | Lifetime | 2.8 | 0.4 | - | - |
| | | | | | | | | | Last 12-month | 1.6 | 0.3 | - | |
| | | | | | | | | | Last month | 0.8 | 0.1 | - | |
| | | | | | | | | | Daily or almost daily | 0.2 | 0.1 | - | |
| 11 | Heydari, 2015 [42] | En | 2012–13 | Fars | Undergraduates of two universities in one city | NR | Mean (SD) Female: 21.2 (2.6) Male: 21.1 (2.1) | 1149 (731; 418) | Lifetime | 4.1 | 3.1 | - | 6 |
| | | | | | | | | | Once in a month | 0.4 | 0.5 | - | |
| | | | | | | | | | Sustained use | 0.5 | 0 | - | |
| 12 | Hakima, 2013 [43] | Pe | 2012 | Ghazvin | Undergraduates of a large governmental non-medical university | NR | Mean: 21.5 Range: 18–40 | 349 (161; 188) | Lifetime | 8.1 | 0.5 | - | 6 |
| 13 | Yaghubi, 2015 [44] | En | 2012 | National | Undergraduates of thirty large governmental non-medical universities | 94.7 | - | 6943 (3200; 3743) | Lifetime | 4.2 | 1.3 | - | - |
| | | | | | | | | | Last 12-month | 2.7 | 1.1 | - | |
| | | | | | | | | | Last month | 1.6 | 0.8 | - | |

(*Continued*)

**Table 2.** (*Continued*)

| | Author, Date | Lang | Year of study | Province | Setting/ Participants | Response rate (%) | Age characteristics (year) | Sample size (Male; Female) | Time indicator | Prevalence of use (%) | | | Numerals of unfulfilled quality items[a] |
|---|---|---|---|---|---|---|---|---|---|---|---|---|---|
| | | | | | | | | | | Male | Female | Total | |
| 14 | Yaghubi, 2017 [45] | Pe | 2012 | National | Undergraduates of thirty large governmental medical universities | 95.9 | - | 3375 (1280; 2095) | Lifetime | 3.3 | 0.9 | - | 2 |
| | | | | | | | | | Last 12-month | 1.3 | 0.5 | - | |
| | | | | | | | | | Last month | 0.9 | 0.2 | - | |
| 15 | Mohammadpoorasl, 2014 [46] | En | 2011 | East Azerbaijan | Undergraduates of nine universities in one city | NR | Mean (SD) 22.1 (2.3) | 1837 (737, 1100) | Lifetime | - | - | 0.6 | 6, 9 |
| 16 | Rezakhani-Moghadam, 2013 [47] | Pe | 2010–11 | Tehran | Students of two large medical and non-medical governmental universities | 97.7 | Mean (SD) TUMS: 22.6 (4.0) TU: 22.9 (3.4) | 977 (452; 525) | Lifetime | 3.8 | 0.8 | - | - |
| 17 | Taremian, 2014 [48][b] | Pe | 2009–10 | Tehran | Undergraduates of three large governmental medical university | 89.5 | - | 3582 (1273; 2309) | Lifetime | 4.2 | 1.2 | - | - |
| | | | | | | | | | Last 12-month | 2.3 | 0.4 | | |
| | | | | | | | | | Last month | 1.4 | 0.2 | | |
| 18 | Amin-esmaeili, 2017 [49][d] | En | 2009 | Tehran | All undergraduates of a large governmental medical university | 90.6 | Mean (SD) 20.1 (1.9) Range: 15–40 | 1541 (508; 1033) | Lifetime | 2.2 | 0.2 | - | - |
| | | | | | | | | | Lifetime (Indirect) | 3.3 | 1.3 | - | |
| | | | | | | | | | Last 12-month | 0.8 | 0.1 | - | |
| | | | | | | | | | Last month | 0.2 | 0.1 | - | |
| | | | | | | | | | Daily or almost daily | 0 | 0 | - | |
| 19 | Amin-esmaeili, 2017 [49][d] | En | 2008 | Tehran | All undergraduates of a large governmental medical university | 90.7 | Mean (SD) 20.2 (1.9) Range: 17–42 | 1660 (561; 1099) | Lifetime | 3.0 | 0.5 | - | - |
| | | | | | | | | | Lifetime (Indirect) | 5.7 | 2.7 | | |
| | | | | | | | | | Last 12-month | 2.3 | 0.3 | - | |

(*Continued*)

**Table 2.** (Continued)

| | Author, Date | Lang | Year of study | Province | Setting/ Participants | Response rate (%) | Age characteristics (year) | Sample size (Male; Female) | Time indicator | Prevalence of use (%) | | | Numerals of unfulfilled quality items[a] |
|---|---|---|---|---|---|---|---|---|---|---|---|---|---|
| | | | | | | | | | | Male | Female | Total | |
| | | | | | | | | | Last month | 1.4 | 0.3 | - | |
| | | | | | | | | | Daily or almost daily | 0.5 | 0 | - | |
| 20 | Amin-esmaeili, 2017 [49][d] | En | 2007 | Tehran | All undergraduates of a large governmental medical university | 96.1 | Mean (SD): 20.2 (2.1) Range: 16–41 | 1633 (591; 1042) | Lifetime | 5.2 | 1.5 | - | - |
| | | | | | | | | | Lifetime (Indirect) | 6.7 | 2.3 | - | |
| | | | | | | | | | Last 12-month | 3.5 | 0.8 | - | |
| | | | | | | | | | Last month | 2.0 | 0.5 | - | |
| | | | | | | | | | Daily or almost daily | 0 | 0.3 | - | |
| 21 | Shams-Alizadeh, 2008 [50] | Pe | 2006–7 | Kurdistan | All undergraduates of a large governmental medical university | 89.0 | Mostly: 20–22 | 1041 (427; 614) | Lifetime | 6.8 | 3.1 | - | - |
| 22 | Sohrabi, 2009 [51] | Pe | 2006–7 | Five provinces [e] | Undergraduate of five large universities of a range of majors | NR | Mostly: 19–25 | 8352 (3372; 4980) | Lifetime | 3.9 | 0.4 | - | 6 |
| 23 | Amin-esmaeili, 2017 [49][d] | En | 2006 | Tehran | All undergraduates of a large governmental medical university | 96.8 | Mean (SD): 20.4 (2.6) Range: 15–43 | 1705 (581; 1124) | Lifetime | 4.5 | 0.7 | - | - |
| | | | | | | | | | Lifetime (Indirect) | 4.7 | 1.0 | - | |
| | | | | | | | | | Last 12-month | 1.7 | 0.4 | - | |
| | | | | | | | | | Last month | 1.0 | 0.2 | - | |
| | | | | | | | | | Daily or almost daily | 0.7 | 0 | - | |
| 24 | Taremian, 2008 [52][b] | Pe | 2005–6 | Tehran | Undergraduate of six large universities of a range of majors | NR | - | 2500 (902; 1598) | Lifetime | 5.2 | 0.6 | - | 6 |
| | | | | | | | | | Last 12-month | 3.2 | 0.3 | - | |
| | | | | | | | | | Last month | 1.9 | 0.2 | - | |

(*Continued*)

**Table 2.** (Continued)

| | Author, Date | Lang | Year of study | Province | Setting/ Participants | Response rate (%) | Age characteristics (year) | Sample size (Male; Female) | Time indicator | Prevalence of use (%) | | | Numerals of unfulfilled quality items[a] |
|---|---|---|---|---|---|---|---|---|---|---|---|---|---|
| | | | | | | | | | | Male | Female | Total | |
| 25 | Zarrabi, 2009 [53] | En | 2005–6 | Guilan | Undergraduates of one large medical governmental university | 98.9 | Mean (SD) 22.1 (3.8) | 827 (295; 532) | Lifetime | - | - | 2.8 | 9 |
| | | | | | | | | | Last month | - | - | 0.4 | |
| 26 | Mortazavi-Moghadam, 2009 [54] | Pe | 2003 | South Khorasan | Undergraduate of three large universities of a range of majors | 87.0 | Mostly: 20–24 | 870 (361; 509) | Lifetime | 1.9 | 0.4 | - | - |
| 27 | Talaei, 2008 [55] | En | 2003 | Khorasan Razavi | All undergraduate of a semi-governmental university, human sciences and agriculture majors | NR | Mostly: 18–24 | 843 (485; 358) | Lifetime | 8.0 | 1.1 | - | 6 |
| 28 | Bahreinian, 2003 [56] | Pe | 2001–02 | Tehran | Undergraduates of one large medical governmental university | NR | Mostly: 20–24 | 565 (181; 384) | Lifetime | 6.6 | 0.3 | - | 6 |
| 29 | Navidi, 2002 [57] | Pe | 2001–02 | Tehran | Medical residents of three large governmental university | 68.3 | UK | 1197 (789; 395) | Lifetime | 7.7 | 0.3 | - | - |
| | | | | | | | | | Last 12-month | - | - | 1.0 | |
| | | | | | | | | | Last month | - | - | 0.7 | |
| | | | | | | | | | Daily | - | - | 0 | |
| 30 | Jodati, 2007 [58] | En | 2001 | East Azerbaijan | Male students living in a dormitory of a large governmental medical university | 79.0 | Mostly: 18–22 | 173 (173; 0) | Last 6-month | 6.4 | - | - | - |
| 31 | Rezaei, 2001[59] | Pe | 1999–2000 | Five provinces | Male undergraduates and postgraduates of six large universities | NR | UK | 1267 (1267; 0) | Lifetime | 7.5 | - | - | 6 |
| | | | | | | | | | Less than once a week | 4.7 | - | - | |
| | | | | | | | | | More than once a week | 1.3 | - | - | |

(*Continued*)

**Table 2.** (Continued)

| | Author, Date | Lang | Year of study | Province | Setting/ Participants | Response rate (%) | Age characteristics (year) | Sample size (Male; Female) | Time indicator | Prevalence of use (%) | | | Numerals of unfulfilled quality items[a] |
|---|---|---|---|---|---|---|---|---|---|---|---|---|---|
| | | | | | | | | | | Male | Female | Total | |
| 32 | Ghanizadeh, 2001 [60] | En | 1999 | Fars | Undergraduates of one large governmental university | 96.8 | Range: 18–31 | 213 (189; 21) | Lifetime | - | - | 12.2 | 9 |
| | | | | | | | | | Last 6 months | - | - | 4.7 | |
| 33 | Mousavi, 2003 [61] | Pe | 1998 | Isfahan | All undergraduates of three universities of a range of majors | 95.8 | - | 735 | Lifetime | - | - | 21.3 | 4, 9 |
| 34 | Ahmadi, 2009 [62] | En | NR | Fars | All undergraduates of dentistry in a large governmental medical university | 78.7 | Mean (SD): 23.0 (4.3) | 236 (150; 86) | Lifetime | 4.7 | 2.3 | - | - |
| | | | | | | | | | Current | 0 | 1.2 | - | |
| 35 | Hajipour, 2002 [63] | Pe | NR | Mazandaran | Undergraduates of one large medical governmental university | 84.5 | UK | 278 (155; 123) | Lifetime | 12.9 | 0 | - | 5 |
| | | | | | | | | | Daily | 5.8 | 0 | - | |
| 36 | Navidi, 1997 [64] | Pe | NR | Tehran | Male medical interns of one university | 90.7 | Mean: 26.9 | 204 (204; 0) | Lifetime | 24.0 | - | - | 2, 5 |
| | | | | | | | | | Less than daily | 8.8 | - | - | |
| | | | | | | | | | Daily | 0 | - | - | |

**a:** 1) The source of sampling was well presented and the sample was representative of the target population. 2) The method of sampling was appropriate (random or census). 3) The sample size was adequate (more than 30). 4) The study subjects and the setting were described in detail. 5) The year of the study was stated. 6) The response rate was provided and it was over 70%. If below 70%, the non-responders were not different from respondents in main demographic characteristics. 7) The condition was measured by valid method. 8) Standard criteria were used for the measurement of the condition. 9) Subgroup analyses for sex, recruitment setting, the definition of use, or time indicator were performed.

**b:** personal communications were made for further data.

**c:** age characteristic provided for the recruited sample.

**d:** This is a repeated cross-sectional study in the years 2006 to 2009.

**e:** Tehran, Isfahan, Kerman, Kermanshah, Khorasan Razavi. **Abbreviations**: En: English; Lang: Language; NA: Not applicable; NR: Not reported; NSU: Network scale-up model; Pe: Persian; PRM: Proxy respondent method; SD: Standard deviation; UK: Unknown.

provinces [27]; each round has been presented separately in the relative table and figure. The recruitment settings of included studies were household, street or public places. In the male subgroup, the pooled prevalence estimates were 7.7% (95%CI: 4.5–11.8; $I^2$ = 97.6%; 6 studies) for lifetime cannabis use, 3.8% (95%CI: 2.2–5.9; $I^2$ = 92.7%; 2 studies) for last-12 month use and 0.8% (95%CI: 0.04–2.1; $I^2$ = 2.7%; 2 studies) for last month or current use (Table 5 and S3 Fig). Among the female subgroup, the corresponding estimates were 0.7% (95%CI: 0.04–1.8; $I^2$

**Table 3. Characteristics and results of studies on the prevalence of cannabis use and use disorder among the school students.**

| | Author, Date | Lang | Year of study | Province | Setting/ Participants | Response rate (%) | Age characteristics (year) | Sample size (Male; Female) | Time indicator | Prevalence of use (%) | | | Numerals of unfulfilled quality items[a] |
|---|---|---|---|---|---|---|---|---|---|---|---|---|---|
| | | | | | | | | | | Male | Female | Total | |
| 1 | Bami, 2020 [65] | En | 2018 | Kerman | 10th to 12th grade students in Bam county | NR | Mean (SD) 16.8 (07) | 600 (300; 300) | Lifetime | 4.4 | 0 | 2.2 | 6 |
| | | | | | | | | | Current | 3.7 | 0 | 1.8 | |
| 2 | Bahramnejad, 2020 [66] | En | 2017 | Kerman | 10th grade students from 80 schools | 93.4 | Median: 15 | 2676 (1269; 1407) | Lifetime | 5.4 | 1.7 | 3.4 | - |
| | | | | | | | | | Current | 3.5 | 1.3 | 2.4 | |
| 3 | Vakili, 2016 [67] | En | 2015–16 | Yazd | Male high-school students in Yazd city | NR | Mostly: 14–15 years | 1020 (1020; 0) | Lifetime | 9.5 | - | - | 6 |
| 4 | Pirdehghan, 2017 [68] | En | 2012–13 | Yazd | 12th grade students | 96.4 | Mean (SD) 17.6 (0.6) Range: 16–22 | 704 (448; 256) | Lifetime | 3.1 | 0 | - | - |
| | | | | | | | | | More than once per lifetime | 1.1 | 0 | - | |
| 5 | Nazarzadeh, 2014 [69] | En | 2011–12 | Ilam | 10th grade students from 75 schools | 94.6 | Mean (SD) 16.3 (0.7) | 1894 (937; 957) | Last month | 7.8 | 1.8 | - | - |
| 6 | Alaee, 2011 [70] | Pe | 2010 | Alborz | 9th to 12th grade students | NR | Mean (SD) 16.5 (1.3) | 445 (207; 238) | Lifetime | 2.4 | 0 | - | 6 |
| 7 | Mohammadpoorasl, 2012 [71] | En | 2010 | East Azerbaijan | 10th grade students | 96.0 | Mean (SD) 15.7 (0.7) Range: 14–19 | 4872 (2093; 2779) | Lifetime | 0.6 | 0.1 | - | 9 |
| 8 | Ghavidel, 2012 [72] | Pe | 2008 | Alborz | 11th grade students | NR | Mean: 17.3 | 400 (204; 196) | Lifetime | - | - | 0.3 | 6, 9 |
| | | | | | | | | | Last 12-month | - | - | 0.3 | |
| | | | | | | | | | Last month | - | - | 0.3 | |
| 9 | Ziaaddini, 2011 [73] | En | 2006–07 | Kerman | 12th grade students | NR | Mean (SD) 17.9 (0.6) | 610 (610; 0) | Lifetime | 6.7 | - | - | 6 |
| 10 | Mohammadkhani, 2012 [74] | Pe | 2005–06 | 9 Provinces | 7th to 12th grade students | 94.7 | Range: 13–18 | 2538 (1283; 1255) | Lifetime | 1.2 | 0 | - | - |
| | | | | | | | | | Last 12-month | 1.0 | 0 | - | |
| | | | | | | | | | Last month | 0.9 | 0 | - | |
| 11 | Najafi, 2007 [75] | Pe | 2005–06 | Guilan | 9th to 12th grade students | 98.8 | Mostly: 15–16 | 1927 (1041; 886) | Lifetime | 3.6 | 0.2 | - | - |
| 12 | Mohammadpoorasl, 2008 [76] | Pe | 2005 | East Azerbaijan | 10th grade male students | 96.9 | Mean (SD) 16.3 (0.9) Range: 15–19 | 1777 (1777; 0) | Lifetime | 0.5 | - | - | - |
| 13 | Najafi, 2005 [77] | Pe | 2004–05 | Guilan | 9th to 12th grade students | 98.3 | Mostly: 14–17 | 1474 (751;723) | Lifetime | 2.3 | 0.4 | - | - |
| 14 | Allahverdipour, 2005 [78] | Pe | 2003 | Tehran | 10th grade students in one district | NR | Range: 15–19 | 189 (189; 0) | Current [b] | 0.5 | - | - | 6 |

(*Continued*)

**Table 3.** (*Continued*)

| | Author, Date | Lang | Year of study | Province | Setting/ Participants | Response rate (%) | Age characteristics (year) | Sample size (Male; Female) | Time indicator | Prevalence of use (%) | | | Numerals of unfulfilled quality items[a] |
|---|---|---|---|---|---|---|---|---|---|---|---|---|---|
| | | | | | | | | | | Male | Female | Total | |
| 15 | Ahmadi, 2004 [79] | En | 2001 | Fars | Male high school students | 94 | Mean (SD) 13.6 (0.7) Range: 12–14 | 470 (470; 0) | Lifetime | 0.2 | - | - | - |
| | | | | | | | | | Current use disorder based on DSM-IV | 0 | - | - | |
| 16 | Ziaaddini, 2006 [80, 81] | Pe | 2000–01 | Kerman | 11th and 12th grade students | 94.8 | NR | 3318 (1945; 1373) | Lifetime | 8.3 | 2.8 | - | - |
| | | | | | | | | | Last month | 4.8 | 1.3 | - | |
| | | | | | | | | | Daily | 3.1 | 0.4 | - | |
| 17 | Ahmadi, 2003 [82] | En | 2000 | Fars | High school students | 94.5 | Mean: 16.6 Range: 13–24 | 397 (197; 200) | Lifetime | 5.6 | 0 | - | - |
| | | | | | | | | | Daily [c] | 1.5 | 0 | - | |
| 18 | Sedigh, 2003 [83] | Pe | UK | National | Grade 8th to 11th students | UK | UK | 7556 (3908; 3646) | Lifetime | - | - | 0.3 | 5, 6, 9 |
| | | | | | | | | | Last month | - | - | 0.2 | |
| | | | | | | | | | Daily | - | - | 0.1 | |

**a:** 1) The source of sampling was well presented and the sample was representative of the target population. 2) The method of sampling was appropriate (random or census). 3) The sample size was adequate (more than 30). 4) The study subjects and the setting were described in detail. 5) The year of the study was stated. 6) The response rate was provided and it was over 70%. If below 70%, the non-responders were not different from respondents in main demographic characteristics. 7) The condition was measured by valid method. 8) Standard criteria were used for the measurement of the condition. 9) Subgroup analyses for sex, recruitment setting, the definition of use, or time indicator were performed.

**b:** positive urine test.

**c:** regular use in last month. **Abbreviations**: En: English; Lang: Language; NA: Not applicable; NR: Not reported; Pe: Persian; SD: Standard deviation; UK: Unknown.

= 97.0%; 5 studies) for lifetime use, 0.2% (95%CI: 0.0–0.7; $I^2$ = 90.0%; 3 studies) for last 12-month use, and 0.3% (95%CI: 0.0–2.1; $I^2$ = 69.9; 2 studies) for last month or current use (Table 5 and S4 Fig). Three studies provided the prevalence of daily or almost daily use with the pooled estimate of 0.5% (95%CI: 0.0–1.7; $I^2$ = 77.8%; 3 studies) in men and 0.0% (95%CI: 0.0–1.1; $I^2$ = 14.2%; 3 studies) in women. No study provided data regarding cannabis use disorder among the young general population.

## University students

Thirty-three studies spanning the years 1998 to 2018 reported on the prevalence of cannabis use among university students with a total sample of 111,600 (44.4% male) (Table 2). Three of these were national studies, two conducted in 2012 and one in 2016, two other studies were conducted in 5 provinces, and the other studies were conducted in thirteen different provinces. One study was a repeated survey in one large medical university in Tehran [49]; each year has been presented separately in the relative table and figure. The mean age of respondents ranged from 20.1 to 23.0 years in different studies. Among male students, the pooled prevalence estimate of cannabis use was 5.7% (95%CI: 4.3–7.3; $I^2$ = 96.9%; 20 studies) for lifetime use, 2.9% (95%CI: 1.8–4.4; $I^2$ = 96.6%; 12 studies) for 12-month use, and 1.7% (95%CI:

**Table 4. Characteristics and results of studies on the prevalence of cannabis use and use disorder among the high-risk populations.**

| | Author, Date | Lang | Year of study | Province | Setting/Participants | RR (%) | Age characteristics (year) | Sample size (Male; Female) | Time indicator | Prevalence of use (%) | | | Numerals of unfulfilled quality items[a] |
|---|---|---|---|---|---|---|---|---|---|---|---|---|---|
| | | | | | | | | | | Male | Female | Total | |
| **People who use drugs** | | | | | | | | | | | | | |
| 1 | Rafiei, 2019 [84] | Pe | 2018 | National | People who use drugs in drug treatment and harm reduction facilities, prisons and public areas | NR | Mean (SD) 36.0 (9.7) | 20051 (18497; 1554) | Lifetime | 41.5 | 38.5 | 41.2 | 6 |
| | | | | | | | | | Less than once a month | - | - | 6.4 | |
| | | | | | | | | | Monthly use [b] | - | - | 6.4 | |
| | | | | | | | | | Weekly use [c] | - | - | 5.1 | |
| | | | | | | | | | Daily use [d] | 11.5 | 15.3 | 11.9 | |
| | | | | | | | | | Current main drug of use | - | - | 12.5 | |
| 2 | Rahimi-Movaghar, Unpublished [85] | NA | 2015–19 | Tehran | People referred for treatment of substance use disorder to a clinic | 100 | Mean (SD) 36.5 (12.2) Median: 34.5 | 988 (921; 67) | Main drug of treatment | 10.9 | 9.0 | - | - |
| 3 | Danesh, 2019 [86] | En | 2015 | Golestan | Clients of opioid maintenance treatment programs from 25 outpatient drug treatment clinics | 94.1 | NR | 701 (656; 45) | Lifetime | - | - | 31.6 | 9 |
| | | | | | | | Mean (SD) 39.2 (11.1) | 478 (448; 30) | Current (urine test) | - | - | 9.7 | |
| 4 | Jamshidi, 2016 [87] | En | 2014–15 | Khuzestan | Treatment seeking individuals in self-referred drug rehabilitation centres | NR | Mean (SD) 38.2 (10.5) | 4400 (4289; 111) | Main drug of treatment | 3.7 | 0.9 | - | 6 |
| 5 | Ghaderi, 2017 [88] | En | 2012–13 | Khorasan Razavi | Patients referred for treatment of opioid dependence based on DSM-IV | NR | Mostly: 30–40 | 260 (140; 120) | Lifetime cannabis dependence based on DSM-IV | 25.7 | 5.0 | - | 6 |
| 6 | Eskandarieh, 2013 [89] | En | 2008 | Tehran | People who inject drugs entered rehabilitation centre for mandatory detoxification | NR | Mean: 28.8 | 402 (386; 14) | Current | - | - | 43.3 | 2, 6, 9 |
| 7 | Dolan, 2011 [90] | En | 2007–08 | Tehran | Female individuals seeking treatment for heroin use disorder | 80.0 | Median 37 | 78 (0; 78) | Lifetime | - | 15.4 | - | 2 |
| 8 | Narenjiha, 2009 [91] | Pe | 2007 | National | People who use drugs in drug treatment and harm reduction facilities, prisons and public areas | NR | Mean (SD): 32.5 (9.6) | 7734 (NR) | Current | - | - | 7.9 | 6, 9 |
| | | | | | | | | 7600 (NR) | Current main drug of use | - | - | 2.0 | |
| 9 | Narenjiha, 2005 [92] | Pe | 2004 | National | People who use drugs in drug treatment and harm reduction facilities, prisons and public areas | NR | Mean (SD): 33.6 (10.48) | 4928 (NR) | Lifetime | - | - | 48.8 | 6, 9 |
| | | | | | | | | 4928 (NR) | Current | - | - | 20.9 | |

*(Continued)*

**Table 4.** (*Continued*)

| | Author, Date | Lang | Year of study | Province | Setting/Participants | RR (%) | Age characteristics (year) | Sample size (Male; Female) | Time indicator | Prevalence of use (%) | | | Numerals of unfulfilled quality items[a] |
|---|---|---|---|---|---|---|---|---|---|---|---|---|---|
| | | | | | | | | | | Male | Female | Total | |
| | | | | | | | | 4925 (NR) | Current main drug of use | - | - | 9.3 | |
| 10 | Razzaghi, 2000 [93] | Pe | 1998–99 | National | People who use drugs in drug treatment and harm reduction facilities, prisons and public areas | NR | Mean: 33.6 | 1472 (1375; 97) | Lifetime | - | - | 47.5 | 6, 9 |
| | | | | | | | | | Last month | - | - | 12.6 | |
| | | | | | | | | | Main drug of use in last month | - | - | 2.1 | |
| **Prisoners** | | | | | | | | | | | | | |
| 11 | Moradi, 2020 [94] | En | 2015 | National | Prisoners | 88.8 | Mostly: >45 | 5508 (5314; 194) | Lifetime | - | - | 3.7 | 9 |
| 12 | SeyedAlinaghi, 2017 [95] | En | 2013–14 | Tehran | Male prisoners at entrance to a prison with positive risk factors for HIV | NR | Mostly: 25–34 | 2860 (2860; 0) | Lifetime | 2.2 | - | - | 6 |
| | | | | | | | | | Current | 0.2 | - | - | |
| 13 | Hamzeloo, 2016 [96] | En | 2012 | Golestan | Prisoners fulfilling ADHD criteria (DSM-IV) among male prisoners | 97.3 | Mean (SD) 31.4 (8.1) | 147 (147; 0) | Lifetime | 1.4 | - | - | 5 |
| 14 | Assari, 2014 [97] | En | 2008 | Six provinces [e] | Adults imprisoned for being involved in fatal vehicle accidents in 7 prisons | NR | Mean (SD) 32.4 (7.9) | 51 (51; 0) | Lifetime | 19.6 | - | - | 6 |
| | | | | | | | | | Last 12-month | 11.8 | - | - | |
| | | | | | | | | | Last month | 7.8 | - | - | |
| | | | | | | | | | Current (urine test) | 15.7 | - | - | |
| 15 | Jalilian, 2013 [98] | Pe | 2007 | Kermanshah | Male prisoners due to rubbery, murder and dispute | 88.7 | Mean (SD) 31.1 (7.8) | 546 (546; 0) | Lifetime | 11.9 | - | - | 2 |
| **Other high-risk groups** | | | | | | | | | | | | | |
| 16 | Khezri, 2020 [99] | En | 2017 | Kerman | Homeless individuals aged 15–29 years from homeless shelters, street outreach sites, and drop-in service centers | NR | Mostly 25–29 years | 202 (109; 93) | Last month | - | - | 8.5 | 6, 9 |
| 17 | Heydari, 2016 [100] | En | 2015 | Fars | Street based; Male motorcycle drivers [f] | NR | Mean (SD) 27.0 (9.3) Range: 16–64 | 414 (414; 0) | Lifetime | 3.6 | - | - | 5, 6 |
| | | | | | | | | | Less than once per week | 1.2 | - | - | |
| | | | | | | | | | More than once per week | 2.4 | - | - | |
| 18 | Mohaqeqi-Kamal, 2019 [101] | En | 2015 | Tehran | Homeless individuals being referred to a large shelter | NR | Mean (SD) 47.4 (11.1) | 193 (193; 0) | Current | 3.6 | - | - | 6 |

(*Continued*)

**Table 4.** (Continued)

| | Author, Date | Lang | Year of study | Province | Setting/Participants | RR (%) | Age characteristics (year) | Sample size (Male; Female) | Time indicator | Prevalence of use (%) | | | Numerals of unfulfilled quality items[a] |
|---|---|---|---|---|---|---|---|---|---|---|---|---|---|
| | | | | | | | | | | Male | Female | Total | |
| 19 | Shokoohi, 2019 [102] | En | 2015 | Thirteen large cities | Female sex workers recruited from public street location through peer efforts and health facilities providing harm reduction services | NR | Mean (SD) 35.6 (8.8) | 1347 (0; 1347) | Last month | - | 2.7 | - | 6 |
| 20 | Maarefvand, 2016 [103] | En | 2014 | Tehran | Long-distance truck drivers from public parking lots | NR | Mean: 36.9 Range: 19–65 | 349 (349,0) | Lifetime | 0.9 | - | - | 2, 6, 8 |
| 21 | Bagheri, 2014 [104] | En | 2012 | Tehran | Individuals aged 18–60 years old being at least 10 days homeless in the last month from 5 voluntary or mandatory shelters of the Municipality of Tehran | NR | Range: 18–60 | 593 (513; 80) | Current | 2.7 | 0 | - | 6 |
| 22 | Ahmadi, 2003 [105] | En | 2000 | Fars | Offspring of people with opioid dependence referred to a treatment centre | NR | Mostly 20–39 | 500 (225; 275) | Lifetime | - | - | 2.6 | 6, 9 |

**a:** 1) The source of sampling was well presented and the sample was representative of the target population. 2) The method of sampling was appropriate (random, census, or multistage method). 3) The sample size was adequate (more than 30). 4) The study subjects and the setting were described in detail. 5) The year of the study was stated. 6) The response rate was provided and it was over 70%. If below 70%, the non-responders were not different from respondents in main demographic characteristics. 7) The condition was measured by valid method. 8) Standard criteria were used for the measurement of the condition. 9) Subgroup analyses for sex, recruitment setting, the definition of use, or time indicator were performed.

**b:** once to three times per month in the last 12 months.

**c:** once to six times per week in the last 12 months.

**d:** in the last 12 months.

**e:** Tehran, East Azarbayjan, Golestan, Sistan and Balouchestan, Yazd, and Kermanshah.

**f:** Based on their presence in a particular area of the city at specific times. **Abbreviations**: En: English; Lang: Language; NA: Not applicable; NR: Not reported; Pe: Persian; SD: Standard deviation.

0.9–2.7; $I^2$ = 95.4%; 9 studies) for last month or current use (Table 5 and S5 Fig). Corresponding estimates were 1.1% (95%CI: 0.6–1.7; $I^2$ = 96.2%; 19 studies), 0.6% (95%CI: 0.3–1.0; $I^2$ = 93.7%; 11 studies), and 0.3% (95%CI: 0.1–0.6; $I^2$ = 89.8%; 9 studies), respectively among female students (Table 5 and S6 Fig). Across all years, 0.4% (95%CI: 0.1–0.7; $I^2$ = 69.1%; 5 studies) of male students and 0.02% (95%CI: 0.0–0.1; $I^2$ = 0.5%; 4 studies) of female students reported daily or almost daily use of cannabis. No study was found on cannabis use disorder among the university students.

## High school students

We found 18 studies spanning years 2000 to 2018 that reported on the prevalence of cannabis use in high school students with a total sample size of 32,867 (56.9% male) (Table 3). One study was conducted nationally, another was conducted in 9 provinces, and the other studies

**Table 5. Pooled prevalence of cannabis use through time among general population, young general population, university students, high school students, and "combined youth groups".**

| Indicator | All years | | | 2000–2005 | | | 2006–2010 | | | 2011–2015 | | | 2016–2020 | | |
|---|---|---|---|---|---|---|---|---|---|---|---|---|---|---|---|
| | P (95% CI) | n [a] | I² | P (95% CI) | n | I² | P (95% CI) | n | I² | P (95% CI) | n | I² | P (95% CI) | n | I² |
| **General population—Male** | | | | | | | | | | | | | | | |
| Lifetime | 2.7 (0.6–6.1) | 5 | 98.0 | 6.6 (0.7–17.4) | 2 | 95.0 | - | - | - | 1.4 (0.1–4.0) | 2 | 92.2 | 0.6 (0.3–1.0) | 1 | - |
| Last 12 months | 1.3 (0.1–3.6) | 2 | 97.0 | - | - | - | - | - | - | 1.3 (0.1–3.6) | 2 | 97.0 | - | - | - |
| Last month or current | 0.8 (0.4–1.2) | 5 | 52.4 | 0.7 (0.3–1.3) | 4 | 63.2 | - | - | - | 1.0 (0.3–2.1) | 1 | - | - | - | - |
| Daily or almost daily | 0.9 (0.5–1.4) | 2 | 0.0 | - | - | - | - | - | - | 0.9 (0.5–1.4) | 2 | 0.0 | - | - | - |
| **General population–Female** | | | | | | | | | | | | | | | |
| Lifetime | 0.3 (0.0–0.7) | 5 | 90.6 | 0.4 (0.0–2.3) | 2 | 89.9 | - | - | - | 0.5 (0.0–2.0) | 2 | 87.4 | 0.0 (0.0–0.04) | 1 | - |
| Last 12 months | 0.2 (0.1–0.3) | 2 | 0.0 | - | - | - | - | - | - | 0.2 (0.1–0.3) | 2 | - | - | - | - |
| Last month or current | 0.1 (0.0–0.3) | 4 | 63.7 | 0.01 (0.0–0.1) | 3 | 0.0 | - | - | - | 0.7 (0.1–1.9) | 1 | - | - | - | - |
| Daily or almost daily | 0.03 (0.0–0.5) | 2 | 68.2 | - | - | - | - | - | - | 0.03 (0.0–0.5) | 2 | 68.2 | - | - | - |
| **Young general population—Male** | | | | | | | | | | | | | | | |
| Lifetime | 7.7 (4.5–11.8) | 6 | 97.6 | 10.2 (1.4–25.5) | 3 | 99.1 | - | - | - | 2.4 (1.0–4.3) | 2 | 0.0 | 8.9 (4.9–14.0) | 1 | - |
| Last 12 months | 3.8 (2.2–5.8) | 2 | 92.7 | - | - | - | - | - | - | 3.0 (1.9–4.2) | 1 | - | 4.1 (2.1–6.7) | 1 | - |
| Last month or current | 0.8 (0.04–2.1) | 2 | 2.7 | 1.5 (0.02–4.6) | 1 | - | - | - | - | 0.5 (0.0–2.0) | 1 | - | - | - | - |
| Daily or almost daily | 0.5 (0.0–1.7) | 3 | 77.8 | - | - | - | - | - | - | 0.6 (0.0–2.7) | 2 | 81.7 | 0.5 (0.0–3.2) | 1 | - |
| **Young general population—Female** | | | | | | | | | | | | | | | |
| Lifetime | 0.7 (0.04–1.8) | 5 | 97.0 | 1.9 (0.1–5.7) | 3 | 97.0 | - | - | - | 1.1 (0.02–3.1) | 1 | - | 0.1 (0.02–0.3) | 1 | - |
| Last 12 months | 0.2 (0.00–0.7) | 3 | 90.1 | - | - | - | - | - | - | 1.4 (0.00–7.0) | 2 | 96.4 | 0.03 (0.0–0.1) | 1 | - |
| Last month or current | 0.3 (0.0–2.1) | 2 | 69.9 | 0.0 (0.0–0.6) | 1 | - | - | - | - | 1.1 (0.02–3.1) | 1 | - | - | - | - |
| Daily or almost daily | 0.0 (0.0–1.1) | 3 | 14.2 | - | - | - | - | - | - | 0.07 (0.0–1.1) | 2 | 71.4 | 0.0 (0.0–0.2) | 1 | - |
| **University students—Male** | | | | | | | | | | | | | | | |
| Lifetime | 5.7 (4.3–7.3) | 20 | 96.9 | 6.0 (4.3–8.0) | 6 | 82.2 | 4.1(3.3–4.9) | 5 | 57.4 | 5.4 (3.5–7.8) | 6 | 90.6 | 10.5 (10.2–10.9) | 3 | 0.0 |
| Last 12 months | 2.9 (1.8–4.4) | 12 | 96.6 | 4.3 (1.7–8.0) | 2 | 72.3 | 1.8 (1.1–2.7) | 2 | 70.1 | 2.0 (1.4–2.7) | 5 | 58.9 | 6.2 (4.4–8.3) | 3 | 84.8 |
| Last month or current | 1.7 (0.9–2.7) | 9 | 95.4 | 1.9 (1.1–2.9) | 1 | - | 1.1 (0.6–1.8) | 2 | 60.9 | 1.2 (0.6–1.8) | 4 | 60.2 | 4.1 (3.6–4.6) | 2 | 9.4 |
| Daily or almost daily | 0.4 (0.1–0.7) | 5 | 69.1 | 1.3 (0.7–2.0) | 1 | - | 0.2 (0.0–0.7) | 1 | - | 0.4 (0.1–0.8) | 2 | 13.4 | 0.3 (0.0–1.1) | 1 | - |
| **University students—Female** | | | | | | | | | | | | | | | |
| Lifetime | 1.1 (0.6–1.7) | 19 | 96.2 | 0.5 (0.3–0.8) | 5 | 0.0 | 0.9 (0.5–1.4) | 5 | 86.1 | 1.2 (0.6–1.9) | 6 | 78.5 | 3.3 (3.1–3.5) | 3 | 0.0 |
| Last 12 months | 0.6 (0.3–1.0) | 11 | 93.7 | 0.3 (0.1–0.6) | 1 | - | 0.3 (0.1–0.5) | 2 | 46.9 | 0.6 (0.24–1.0) | 5 | 74.1 | 1.3 (0.4–2.5) | 3 | 86.8 |
| Last month or current | 0.3 (0.1–0.6) | 9 | 89.8 | 0.2 (0.02–0.5) | 1 | - | 0.2 (0.1–0.4) | 2 | 0.0 | 0.3 (0.02–0.8) | 4 | 81.9 | 1.0 (0.9–1.2) | 2 | 0.0 |
| Daily or almost daily | 0.02 (0.0–0.1) | 4 | 0.5 | - | - | - | 0.03 (0.0–0.2) | 1 | - | 0.03 (0.0–0.2) | 2 | 0.0 | 0.0 (0.0–0.3) | 1 | - |
| **High school students—Male** | | | | | | | | | | | | | | | |
| Lifetime | 3.2 (1.7–5.3) | 14 | 97.0 | 2.4 (0.6–5.3) | 7 | 97.5 | 2.7 (0.1–8.5) | 3 | 96.9 | 6.0 (1.3–13.7) | 2 | 95.5 | 5.1 (4.1–6.3) | 2 | 0.0 |
| Last 12 months | 1.0 (0.5–1.6) | 1 | - | 1.0 (0.5–1.6) | 1 | - | - | - | - | - | - | - | - | - | - |
| Last month or current | 3.2 (1.4–5.6) | 6 | 94.7 | 1.8 (0.05–5.4) | 3 | 96.3 | - | - | - | 7.8 (6.2–9.6) | 1 | - | 3.5 (2.7–4.5) | 2 | 0.0 |
| Daily or almost daily | 2.7 (1.6–3.9) | 2 | 28.7 | 2.7 (1.6–4.0) | 2 | 28.7 | - | - | - | - | - | - | - | - | - |
| **High school students—Female** | | | | | | | | | | | | | | | |

*(Continued)*

**Table 5.** (Continued)

| Indicator | All years | | | 2000–2005 | | | 2006–2010 | | | 2011–2015 | | | 2016–2020 | | |
|---|---|---|---|---|---|---|---|---|---|---|---|---|---|---|---|
| | P (95% CI) | n [a] | $I^2$ | P (95% CI) | n | $I^2$ | P (95% CI) | n | $I^2$ | P (95% CI) | n | $I^2$ | P (95% CI) | n | $I^2$ |
| Lifetime | 0.3 (0.0–0.9) | 10 | 91.8 | 0.4 (0.0–1.7) | 5 | 94.4 | 0.1 (0.0–0.2) | 2 | 0.0 | 0.0 (0.0–0.7) | 1 | - | 0.6 (0.0–3.3) | 2 | 90.6 |
| Last 12 months | 0.00 (0.0–0.1) | 1 | - | 0.00 (0.0–0.1) | 1 | - | - | - | - | - | - | - | - | - | - |
| Last month or current | 0.6 (0.1–1.7) | 5 | 91.8 | 0.4 (0.0–2.7) | 2 | 96.3 | - | - | - | 1.8 (1.0–2.7) | 1 | - | 0.5 (0.0–2.4) | 2 | 86.4 |
| Daily or almost daily | 0.3 (0.05–0.7) | 2 | 0.0 | 0.3 (0.05–0.7) | 2 | 0.0 | - | - | - | - | - | - | - | - | - |
| **Combined youth groups—Male** | | | | | | | | | | | | | | | |
| Lifetime | 5.2 (4.1–6.6) | 40 | 97.8 | 4.8 (2.7–7.5) | 16 | 97.9 | 3.7 (2.4–5.2) | 8 | 92.5 | 5.0 (3.4–7.0) | 10 | 91.1 | 8.5 (6.5–10.9) | 6 | 95.2 |
| Last 12 months | 3.0 (2.0–4.1) | 15 | 96.4 | 2.3 (0.7–4.7) | 3 | 87.3 | 2.0 (1.3–2.9) | 2 | 63.5 | 2.2 (1.6–2.8) | 6 | 56.4 | 4.9 (3.4–6.7) | 4 | 94.8 |
| Last month or current | 2.0 (1.3–2.9) | 17 | 94.5 | 1.8 (0.4–3.8) | 5 | 92.9 | 1.1 (0.6–1.8) | 2 | 60.9 | 2.0 (0.6–4.1) | 6 | 95.0 | 4.0 (3.8–4.2) | 4 | 0.0 |
| Daily or almost daily | 0.6 (0.2–1.1) | 10 | 87.1 | 1.9 (0.8–3.6) | 3 | 83.9 | 0.2 (0.0–0.7) | 1 | - | 0.5 (0.05–1.2) | 4 | 71.4 | 0.4 (0.0–1.5) | 2 | 69.8 |
| **Combined youth groups—Female** | | | | | | | | | | | | | | | |
| Lifetime | 0.8 (0.4–1.2) | 37 | 96.6 | 0.7 (0.2–1.5) | 13 | 94.3 | 0.7 (0.3–1.1) | 7 | 87.4 | 1.0 (0.5–1.6) | 8 | 75.8 | 0.8 (0.1–2.2) | 6 | 98.4 |
| Last 12 months | 0.4 (0.2–0.8) | 15 | 95.6 | 0.1 (0.0–0.5) | 2 | 77.8 | 0.3 (0.1–0.5) | 2 | 46.9 | 0.7 (0.3–1.3) | 7 | 86.1 | 0.3 (0.0–1.3) | 4 | 97.8 |
| Last month or current | 0.4 (0.2–0.7) | 16 | 88.8 | 0.2 (0.0–1.0) | 4 | 90.0 | 0.2 (0.1–0.4) | 2 | 0.0 | 0.6 (0.2–1.2) | 6 | 84.9 | 0.9 (0.5–1.4) | 4 | 59.9 |
| Daily or almost daily | 0.03 (0.0–0.1) | 9 | 34.4 | 0.3 (0.05–0.7) | 2 | 0.0 | 0.03 (0.0–0.2) | 1 | - | 0.01 (0.0–0.1) | 4 | 18.8 | 0.0 (0.0–0.1) | 2 | 0.0 |

**a:** Number of studies.

were conducted in eight different provinces. The mean ages of respondents ranged from 13.6 to 17.9 years. The pooled estimates of lifetime prevalence of cannabis use were 3.2% (95%CI: 1.7–5.3; $I^2$ = 97.0%; 14 studies) and 0.3% (95%CI: 0.0–0.9; $I^2$ = 91.8%; 10 studies) in male and female students, respectively. Only one study reported on the last 12-month prevalence among high-school students, 1.0% of male and 0.0% of female students report such use [74]. The pooled prevalence of last month or current use were 3.2% (95%CI: 1.4–5.6; 6 studies) among male students (Table 5 and S7 Fig) and 0.6% (95%CI: 0.1–1.7; $I^2$ = 91.8%; 5 studies) among female students (Table 5 and S8 Fig). The corresponding estimates for daily or almost daily use were 2.7% (95%CI: 1.6–3.9; $I^2$ = 28.7; 2 studies) and 0.3% (95%CI: 0.05–0.7; $I^2$ = 0.0%; 2 studies) among male and female students, respectively. Cannabis use disorder was assessed in only in one study. Conducted in 2001 among male students in one province, no current use disorder was detected among the students.

## High-risk groups

In total, 22 studies reported on cannabis use among high-risk population groups in Iran (Table 4). We categorized these studies based on their target population into PWUD (10 studies), prisoners (5 studies), and other high-risk groups (7 studies).

**People who use drugs.** We found 7 studies spanning years 1998 to 2018 reporting the prevalence of cannabis use among PWUD with a total sample size of 35,366. Four of these were repeated national situation assessment surveys conducted in 1998, 2004, 2007 and 2018. S9 Fig presents the prevalence of cannabis use among PWUD based on timeframe and frequency. The pooled estimate of lifetime and last month or current cannabis use were 38.8% (33.3–44.4; $I^2$ = 97.7%; 5 studies) and 16.2% (11.3–21.8; $I^2$ = 99.2%; 6 studies), respectively.

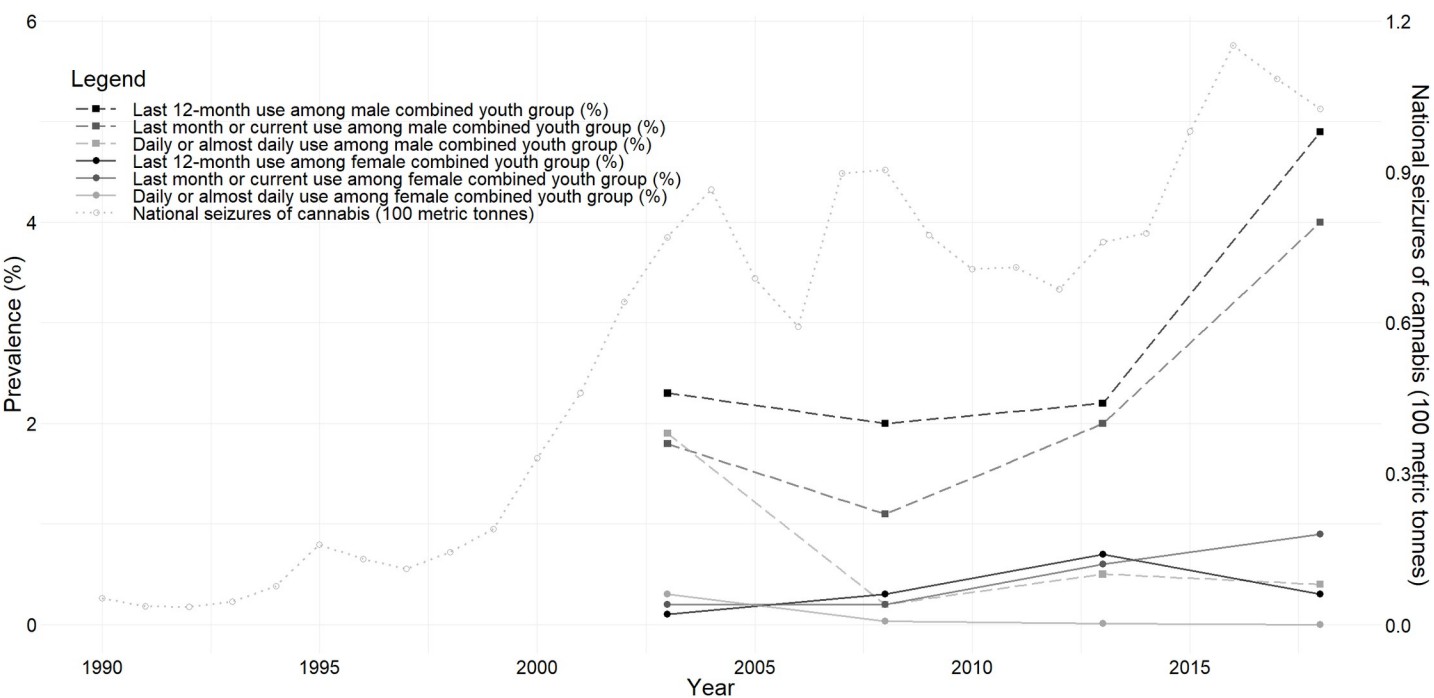

**Fig 2. Time trend of cannabis use among "combined youth group" and national seizures of cannabis (100 metric tonnes- both resin and plant forms).** The cannabis use among male and female of "combined youth groups" were pooled for 2000–2005, 2006–2010, 2011–2015, and 2016–2020 periods and were plotted in the middle of each period.

Only the latest national survey conducted in 2018 assessed the prevalence of last 12-month and daily use of cannabis, estimated at 29.8% and 11.9%, respectively [84].

The four national situation assessment surveys have assessed the prevalence of cannabis being the current main drug of use among the PWUD (S9 Fig). The corresponding figure was 12.5% in the latest study in 2018. Three other studies reported on treatment-seeking and treatment referral for cannabis use among PWUD. The results of these studies are not presented in the forest plot. Two of these studies reported on treatment-seeking for cannabis use. One recruited 988 PWUD (93.2% male) referred for treatment in 2015–19 from one treatment centre. Of these, 10.9% of men and 9.0% of women sought treatment for cannabis use disorder [85]. The other study recruited 4400 individuals from drug rehabilitation centres in 2014–15, 3.7% of male and 0.9% of female clients were referred for cannabis use disorder [87]. A third study assessed lifetime cannabis dependence (based on DSM-IV) among patients referred for treatment of opioid dependence; 25.7% of the male patients and 5.0% of the female patients met these criteria [88].

**Prisoners.** Five studies spanning years 2007 to 2015 examined the prevalence of cannabis use in a total sample of 9,112 prisoners (97.9% male). One study was conducted nationally, another was conducted in 6 provinces, and the other three were conducted in three different provinces. The pooled lifetime prevalence of cannabis use in these studies was 5.4% (95%CI: 2.7–8.8; I$^2$ = 96.0%; 5 studies; S10 Fig). Current use of cannabis was reported in 0.2% male prisoners in one study and 15.7% in the other. No study evaluated cannabis use disorder among the prisoners.

**Other high-risk groups.** Three studies were conducted among homeless individuals [99, 101, 104]. One only recruited homeless individuals aged between 15–29 years, reported 8.5% of total sample (N = 202) had used cannabis in the last month [99]. In the other two studies,

3.6% (N = 193) and 2.7% (N = 513) of male homeless people and no one (N = 80) among female subgroup reported current cannabis use [101, 104].

The other four studies were conducted among other high-risk subgroups (Table 4). One study conducted among female sex workers in 13 large cities in the country in 2015 (N = 1347), reported a 2.7% prevalence of cannabis use in the last month [102]. One other study reported that among male motorcycle drivers (N = 414) in a large city, 2.4% had used cannabis more than once per week in lifetime [100]. Another study conducted among male long-distance truck drivers (N = 349) reported a lifetime prevalence of cannabis use of 0.9% [103]. Finally, the seventh study recruited offspring of people with opioid dependence referred to a treatment centre (N = 500; 45% male; mostly 20–29 years) [105]. Of these, 2.6% of the offspring reported lifetime cannabis use.

### Heterogeneity study

The meta-regression analysis showed that the prevalence of cannabis use was significantly higher in males compared to females (p<0.001), high-risk population compared to the general population (p<0.001), and young age group to the general population (p = 0.01). Prevalence estimates of last 12-month and daily and almost daily use were lower compared to lifetime use (p = 0.03 and 0.006 respectively). Other variables (i.e. study year, number of unfulfilled quality criteria and other participant groups) showed no significant association with cannabis use prevalence (S3 Table).

### Trend

Fig 2 and S4 Table present the trends in the prevalence of cannabis use according to timeframe and frequency of use. To evaluate changes in cannabis use over time, we pooled data from all studies conducted in youth. Sixty studies reported on the prevalence of cannabis use among youths (nine in the young general population, 33 in university students, and 18 in high school students).

Among males in this combined sample of studies, the prevalence of last 12-month use of cannabis increased significantly from 2000 to 2020 (b = 0.05; $P$ = 0.035). The last 12-month prevalence was 2.3% (95%CI: 0.7–4.7%; $I^2$ = 87.3%; 3 studies) in years before 2005 and reached 4.9% (95%CI: 3.4–6.7; $I^2$ = 94.8%; 4 studies) in the 2016–2020 period. The linear trends in the lifetime or last month or current prevalence were not significant (S4 Table). Among females, the prevalence estimate did not change for any timeframes (S4 Table). The pooled estimates of cannabis use in different periods based on sex subgroups are shown in Table 5. The time trend was somewhat different in university student samples. While there was no significant trend in the prevalence of last 12-month cannabis use among male university students (b = 0.004; $P$ = 0.3); the linear trend of last 12-month cannabis use among female university students showed a significant increase from 2000 to 2020 (b = 0.005; $P$ = 0.02).

The time trend of the prevalence of last 12-month use of cannabis among the "combined youth group" in Tehran province was investigated, as well. The linear trends were significant both in the males (b = 0.01; $P$ = 0.01) and female (b = 0.008; $P$ = 0.006) subgroups. It should be noted than except one, the other studies in Tehran were conducted among the university students.

### Geographical distribution

S11 Fig shows the pooled prevalence of last 12-month cannabis use in male and female in the combined youth group in six provinces in Iran. No data were available at province level for 25 other provinces. The highest prevalence in the male combined youth group was reported in

the Fars province (7.3%; one study). Whereas, the highest prevalence in the female combined youth group was reported from Tehran province (0.6%; six studies).

## Quality assessment

The number of unfulfilled quality items for all studies is presented in Tables 1–4. Among the 90 studies, there were only six with three unfulfilled items out of the nine. No study had more than three unfulfilled quality items. With the removal of one study in the young female general population [28], the pooled estimate of last 12-month cannabis use among the young female general population was reduced from 0.2% (95%CI: 0.0–0.7) to 0.04% (95%CI: 0.0–1.3). With the removal of another study among PWUD [89], the pooled estimate of last month or current use in this population changed from 16.2% (95%CI: 11.3–21.8) to 12.2% (95%CI: 8.3–16.8). Removal of the study among the male general population from the meta-analysis changed the pooled estimate of last month or current use less than 0.1% [25]. The other three studies were not included in the meta-analysis [19, 83, 103].

## Discussion

The current study is the first systematic review in Iran to provide an estimate of various cannabis use indicators–i.e., lifetime, last 12-month, last month or current, and daily or almost daily use—among the general population and high-risk population, in addition to the youths. In addition, this is the first review on the prevalence of cannabis use disorder in Iran. Due to the extensive search applied in this study, we could successfully retrieve 50 studies with high quality not identified from the online databases. The previous systematic review conducted up to 2014 on the lifetime cannabis use [9], including a total of 33 studies had supporting results, 4.0% among Iranian university and high-school students with higher rates among university and male students, in the similar study span for the current study.

We found that in Iran, 1.3% of the male general population and 2 per 1000 of the female general population used cannabis in the last 12-month. The overall prevalence is around 0.8% for the general population of the country. These estimates are based on the most recent national surveys conducted in 2011 and 2015. The pattern of cannabis use among sex subgroups is similar to other illicit substances in the Iranian population. The prevalence of cannabis use in the general population is lower than the prevalence of soft opioid use (such as opium at 4.4%) and higher than hard opioids (such as heroin at 0.4%) or stimulants (at less than 0.5%) [106, 107].

The United Nation Office on Drugs and Crime (UNODC) estimates that the last 12-month prevalence of cannabis use in the general population aged 15–64 was about 3.9% in 2019 globally [1], five times higher than in Iran. Notably, the prevalence of cannabis use in Iran is lower than the other countries in the region such as Pakistan (3.6%) [108], Egypt (6.2%) [2], Tunisia (2.6%) [2], and Afghanistan (current cannabis use: 3.8%) [109]. The estimates of last 12-month use are also higher in India (3.0%) [110], in African countries (6.4%) [110], in the European Union countries (7.4%) [111], and in Australia (10.4%) [112], making cannabis the most prevalent substance used in many of these countries. The annual prevalence of cannabis use is much higher in Uruguay (15.0%) [113], Canada (15.0%) [114], and the USA (15.9%) [115] where the use of cannabis is partially legalized.

After pooling data for the combined youth groups, we found higher 12-month prevalence estimates for the most recent period (2016–2020)– 4.9% among males, 0.3% among females, and 2.6% in the total combined youth group. Based on the latest national census in Iran, we estimate that 745,000 Iranians aged 15–34 years use cannabis annually. These estimates are higher than the general population prevalence estimates. A similar age pattern in the

prevalence of cannabis use has been noted in other countries [111, 114, 115]. Furthermore, the female to male ratio among Iranian youth in the 2016–2020 period was higher than the earliest period (1/16 vs. 1/23), suggesting the increasing popularity of cannabis use among young Iranian females. However, female to male ratio among Iranian youth is still much lower than in European Union countries (1/2) [111] and the USA (1/1.2) [115].

Similar to the general population estimates, the prevalence of cannabis use among Iranian youth (2.6%) is lower than youths in many other countries. The last year prevalence of cannabis use in most of the European Union countries is approximately 20% among the population aged 15–24 years, which is almost five times higher than the prevalence of use of other illicit drugs combined and also higher than the prevalence in the 15-64-year-old general population (7.4%) [111]. The 12-month prevalence estimates are similarly high among youth in other industrialized countries: e.g., 19% among 15–19 years old and 33% among 20–24 years old Canadians [114], and 34.8% among 18–25 years old in the USA [115]. Limited data is available on cannabis use among the young population of Eastern Mediterranean region countries.

The data on the prevalence of cannabis use disorder is consistent with international data in showing a lower prevalence in Iran compared to other countries. According to the latest national survey, 0.5% of the Iranian general population aged 15–64 met the criteria for cannabis use disorder in the last 12-month [21]. While higher than the global estimate of the prevalence of cannabis use disorder (0.3%) [116], the rate in Iran is less than many other countries, including the US, European countries and India [111, 115, 117]. This pattern is also reflected in treatment-seeking for cannabis use disorder. A total of 3.6% and 10.9% of all clients seeking treatment for substance use disorder in two different provinces in Iran, sought treatment for cannabis use disorder. The majority was male (95%) with a mean age of 36 years. The pattern is somehow different from industrialized countries, where a higher percentage of cannabis use disorder is seen among those admitted for drug abuse treatment, with a younger age at admission and a larger proportion of females [111, 118].

We found an increasing trend of last 12-month cannabis use among male youth between 2000 and 2020. No significant trends were found among female youth. However, we found some evidence on an increase in cannabis use among female university students. Furthermore, there was significant increasing trend among youths (the majority being university students) in Tehran province in male and female subgroups. The observed increases are in line with the significant rise of national cannabis seizures. The amount of total cannabis seizures has increased significantly from 1990 to 2018 in Iran. Resin constituted the main form of seizures. There were reported of cannabis plant seizures only in the years 2000, and 2007 to 2011, constituting less than 20% of the total annual cannabis seizure in these years. The cannabis seized in Iran has been reported to be imported from Afghanistan and Pakistan, making Iran a transit country for cannabis. It has been reported that only 20% of the cannabis entering Iran was for domestic use, 65% destined for the Arabian Peninsula and 15% destined for Caucasus [1]. Cannabis resin seized in Afghanistan and Pakistan as two of the main cannabis resin producing countries has also been increasing for more than two decades [14]. There are no precise data on the extent of cannabis cultivation inside Iran, although there are reports of discovery and destruction of indoor and outdoor grown plants and farms.

The observed trend in Iran may also be linked to the legalization of medicinal and recreational use of cannabis in several countries [119]. While cannabis is categorized as a controlled substance (Schedules I) internationally, some countries have changed or are perusing change in the level of cannabis control and related legislations [6, 120–122]. According to drug control law in Iran, the use of cannabis is illegal and cannabis is categorized in the same control level as opium, but lower than heroin, cocaine, and methamphetamine. Nevertheless, learning about the highly publicized changes in cannabis policy in the USA and other countries may

have impacted attitudes of the Iranian youth toward harms associated with cannabis use [6]. The growing global prevalence of cannabis use in the last two decades [1] in conjunction with the legalization trends in several industrialized countries has raised concerns about exposure of youth to the potentially harmful effects of cannabis [1, 122, 123].

Cannabis use, especially frequent use might be associated with various short-term and long-term health outcomes [5, 6, 123, 124]. Cannabis use disorder is one of the main associated harms [5], which itself is a strong predictor of negative health outcomes [125]. Chronic psychotic disorders and depression in individuals with predisposing factors have been linked to cannabis use with a dose-response relationship [126, 127]. Early and regular use of cannabis impairs the development of the brain and negatively affect the educational outcomes [5]. Furthermore, cannabis use impairs driving skills and result in a modest increase in the risk of car accident [128–130]. Health consequences of cannabis use in Iran have not been extensively assessed. There are some reports on cases of cannabis-related poisoning cases referred to hospitals in different provinces in Iran, accounting for 1% to 2% of all admitted drug poisoning cases [131–133], including unintentional pediatric poisoning cases [134]. It can be anticipated that with the increase in cannabis use, especially in youth, the adverse health effects might arise. Although the precise effects of the changes in cannabis demand and supply on public health remain unexplored, education of the public, health experts, and policymakers on the cannabis adverse health outcomes and the possible negative effect of cannabis is important [123].

## Limitations

In interpreting the study results, several limitations should be considered. First, we did not find recent studies among the general population which provided data on the main indicators of cannabis use in the last 5 years. Furthermore, because of the inadequate number of studies in each period, the trend plot was not presented for the general population. Due to the same limitation, studies conducted among the young general population, university students, and high school students were merged to form a combined youth group for the trend analysis. Second, although some of the studies did not report whether daily or almost daily use indicator was in the lifetime, last 12-month or last month, we decided to pool them into a "daily use in the last month or current", due to small numbers of studies. Furthermore, we pooled data on last month use with current use due to the scarcity of studies reporting these measures. Third, there were no separate prevalence data for the combined youth group for 25 out of 31 provinces of the country to investigate the possible differences in various regions and the trend in other provinces other than Tehran. Fourth, it should be noted that the estimates might be under-reported as cannabis use is illegal. Also, recall bias would affect the estimated prevalence. Fifth, due to the high heterogeneity, the results should be interpreted with caution. Sixth, further studies are required to better elucidate the extent of cannabis use disorder and treatment seeking in the country. Finally, due to multiple sources approached for accessing all possible relevant studies, we could not track the numbers in the stages of the screening process for the 50 studies in the opportunistic methods.

## Conclusion

In the context of the limitations noted above, this study provides the first overview of cannabis use and use disorder prevalence in the country. The prevalence of cannabis use in Iran appears to be lower than the prevalence in many other countries. However, along with the increase in cannabis seizures, there is strong evidence of an increase in cannabis use among the youth. Moreover, there is some evidence of an increase in cannabis use disorder. There is a need to

monitor cannabis use and the perception of associated risks in the national population and various subgroups, especially among the youth. Moreover, preventive and educational programs in schools and out of schools are needed.

## Supporting information

**S1 Fig. The pooled prevalence of cannabis use among the male general population.**
(DOCX)

**S2 Fig. The pooled prevalence of cannabis use among the female general population.**
(DOCX)

**S3 Fig. The pooled prevalence of cannabis use among the male young general population.**
(DOCX)

**S4 Fig. The pooled prevalence of cannabis use among the female young general population.**
(DOCX)

**S5 Fig. The pooled prevalence of cannabis use among male university students.**
(DOCX)

**S6 Fig. The pooled prevalence of cannabis use among female university student.**
(DOCX)

**S7 Fig. The pooled prevalence of cannabis use among male school students.**
(DOCX)

**S8 Fig. The pooled prevalence of cannabis use among female school students.**
(DOCX)

**S9 Fig. The pooled prevalence of cannabis use among people who use drugs.**
(DOCX)

**S10 Fig. The pooled prevalence of lifetime cannabis use among male prisoners.**
(DOCX)

**S11 Fig.** The pooled prevalence of last 12-month cannabis use among "combined youth groups" in different provinces; a) male subgroup b) female subgroup. The numbers on each province are the pooled estimates and the numbers in the parenthesis are the number of studies.
(DOCX)

**S1 Table. Search strategies used in international databases.**
(DOCX)

**S2 Table. Quality assessment tools.**
(DOCX)

**S3 Table. Meta-regression of possible sources of heterogeneity.**
(DOCX)

**S4 Table. Trends of various cannabis use measures among the "combined youth groups" and national cannabis seizures.**
(DOCX)

**S1 Checklist. PRISMA checklist.**
(DOCX)

## Acknowledgments

We would like to extend our appreciation Dr. Nouzar Nakhaei, Dr. Zaher Kahzaei, Dr. Hamid Yaghubi, Dr. Mohammad Hamzeloo, and Dr. Ali Mirzazadeh for providing further data and analysis.

## Author Contributions

**Conceptualization:** Masoumeh Amin-Esmaeili, Afarin Rahimi-Movaghar.

**Data curation:** Yasna Rostam-Abadi, Masoumeh Amin-Esmaeili, Shahab Baheshmat, Marziyeh Hamzehzadeh, Hossein Rafiemanesh, Anousheh Safarcherati, Farhad Taremian.

**Formal analysis:** Yasna Rostam-Abadi, Jaleh Gholami, Shahab Baheshmat, Afarin Rahimi-Movaghar.

**Funding acquisition:** Afarin Rahimi-Movaghar.

**Methodology:** Yasna Rostam-Abadi, Jaleh Gholami, Masoumeh Amin-Esmaeili, Morteza Nasserbakht, Leila Ghalichi, Ramin Mojtabai, Afarin Rahimi-Movaghar.

**Project administration:** Anousheh Safarcherati.

**Supervision:** Afarin Rahimi-Movaghar.

**Validation:** Ramin Mojtabai, Afarin Rahimi-Movaghar.

**Writing – original draft:** Yasna Rostam-Abadi, Shahab Baheshmat, Marziyeh Hamzehzadeh, Afarin Rahimi-Movaghar.

**Writing – review & editing:** Yasna Rostam-Abadi, Jaleh Gholami, Masoumeh Amin-Esmaeili, Hossein Rafiemanesh, Morteza Nasserbakht, Leila Ghalichi, Anousheh Safarcherati, Farhad Taremian, Ramin Mojtabai, Afarin Rahimi-Movaghar.

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
