## [Decision Letter · Decision Letter 0]

18 Jun 2021

PONE-D-21-10629

Evidence for an increase in cannabis use in Iran – A systematic review and trend analysis

PLOS ONE

Dear Dr. Rahimi-Movaghar,

Thank you for submitting your manuscript to PLOS ONE. After careful consideration, we feel that it has merit but does not fully meet PLOS ONE’s publication criteria as it currently stands. Therefore, we invite you to submit a revised version of the manuscript that addresses the points raised during the review process.

We look forward to receiving your revised manuscript.

Kind regards,

Chaisiri Angkurawaranon

Academic Editor

PLOS ONE

Journal Requirements:

2. In the Methods section please provide additional information regarding the background and training of the experts consulted during the literature search.

Furthermore, please provide additional details regarding the validation of the quality assessment tool used.

Finally please provide additional details regarding how cannabis use disorder was defined as a part of the study inclusion criteria.

Reviewers' comments:

Reviewer's Responses to Questions

**Comments to the Author**

1. Is the manuscript technically sound, and do the data support the conclusions?

Reviewer #1: Yes

Reviewer #2: Partly

2. Has the statistical analysis been performed appropriately and rigorously? 

Reviewer #1: Yes

Reviewer #2: Yes

3. Have the authors made all data underlying the findings in their manuscript fully available?

Reviewer #1: Yes

Reviewer #2: No

4. Is the manuscript presented in an intelligible fashion and written in standard English?

Reviewer #1: Yes

Reviewer #2: Yes

5. Review Comments to the Author

Reviewer #1: Methods

- Please provide the list of reviewers in the “Eligibility criteria and screening” section (page 4).

- FigureS1; there were 307 excluded papers (=88%). That is the big number. Thus, the authors should provide the reasons (e.g., not quantitative empirical, no specific effect measure, not primary outcome of interest, or no full text) and also how many papers in each reason.

Results

- Table 1: typo error “Mostly 21-0 years” – Damari, 2020

- Fig S1-S11, the authors did the subgroup analysis, including by gender (male/female), frequency (lifetime, 12-mo, last month,…), university student, et al. However, high heterogeneity has still been found. Testing cause of heterogeneity according to the variation of quality of included studies should be concerned in this study.

- The effect of spatiotemporal (i.e., place and time) will affect the pooled prevalence during 1990 to 2021. Table 5 can explain the temporal effect, but not for spatial effect. In my point of view, the subgroup analysis by regions should be done in this study.

- The authors tried to analyze the pooled prevalence during 1990 to 2021. In fact, the prevalence has been changed year by year. Thus, subgroup analysis by study year might be provide some information to the authors.

Reviewer #2: Dear authors,

A similar systematic review was published by Nazarzadeh et al., (2015). Prevalence of Cannabis Lifetime Use in Iranian High School and College Students: A Systematic Review, Meta-Analyses, and Meta-Regression. DOI: 10.1177/1557988314546667. In my opinion, it would be useful to comment on the added value of this review and to compare your results with the results of the mentioned review that searched for references between 1979 and 2014. Please find below my suggestions to increase the accuracy of reporting. In my opinion, a re-categorization of the used groups and including cannabis use/dependence/cannabis use disorders as a separate outcome could increase the value of the manuscript.

Introduction:

• Line 33-Please clarify the statement “However, there is anecdotal evidence that cannabis use is increasing in the country and is becoming an important public health problem”

• Lines 36-38- Youth is generally (e.g., by the United Nations) defined as 15-24 years and include high school and many university students. Therefore, it would be useful to revisit the three mentioned categories: youth, high-school students, and university students.

• The authors did not provide the rationale neither for studying the prevalence of cannabis use in high risk groups nor for national seizures.

Methods:

• Please specify if the systematic review was registered on the International Prospective Register of Systematic Reviews (PROSPERO) or another repository

• Lines 42- On the world scene, the landscape has significantly changed in the last 2 decades with the legalization and decriminalization of cannabis use. Therefore, it would be useful to provide a rationale for including in the search references starting 1990.

• I recommend that the authors provide their research question(s)

• According to the PRISMA guidelines, it is highly recommended (and necessary) to provide the study eligibility criteria in PICO format. A clear definition of the outcome appears only in the results section, i.e., the authors combined use prevalence with cannabis abuse/dependence/cannabis use disorder (CUD). In my opinion, an important (secondary) outcome would be the prevalence of cannabis abuse/dependence/CUD among cannabis users in general and among frequent users.

• The selected quality appraisal tool is adequate for observational studies of prevalence. The authors mentioned that they included studies of any methodology and design; in my opinion, it would be useful to report how the quality of intervention studies was appraised. If intervention studies were not included, this should be stated in the eligibility criteria

• One of the outcomes of interest was cannabis abuse/dependence/CUD. It would be important that the authors provide additional details on how this outcome was operationalised and how many of the included studies met criterion 6 of the Joanna Briggs Institute appraisal tool “Were valid methods used for the identification of the condition?”

• Line 57, please include the initials of the persons involved in the screening of references and the initials of the person who mediated disagreements

• Lines 60-64, the authors mention that data related to prevalence use was extracted. I am unsure whether this includes cannabis abuse

• Line 71- studies who reported result separate by gender were included in the meta-analyses. What happened with studies who did not report separately by sex or gender? Was this an exclusion criterion?

• Line 73, 80, I recommend that the authors provide a clear definition of the population subgroups; they used a mix of age and education status (students) e.g., what is the difference between young general population and general population? I suggest using groups based on relevant age-ranges (e.g., youth) as a primary outcome and high-risk groups (considering the relative low number of studies, the categories could be collapsed) as a secondary outcome.

• Lines 74-75 Additional details related to frequency of use are needed e.g., how is “currently the main drug” indicative of the frequency of use; what is included in last month or current?

• Line 83: Please explain the meaning of “network scale-up method”

• Lines 87-89, If not enough data was available for some periods, I suggest collapsing categories e.g., 2000-2010. As no rationale was provided for selecting the 5-year time intervals, using 10-years intervals could be a viable alternative. Why was the interval 1990-2000 not used? The same observation applies to the prevalence of cannabis use.

• Lines 95-97, presenting data on national seizure of cannabis is interesting but it is not part of the main objectives, not sure why it was mentioned in the abstract

• Line 98, please provide the name of the package used in R for meta-analyses

Results

• It is common practice to provide the PRISMA flow diagram in the main manuscript (not as an appendix). How can authors explain that more than half (50 out of 90) of included studies were identified by using additional resources (e.g., contacting experts). I recommend that for these additional studies, the authors report how many were initially recommended/identified and how many were excluded at each stage of reference screening (i.e., title and abstract and full text screening stages). I recommend that the authors report the proportion of published studies out of these 50 additional references and the results of quality appraisal.

• I recommend that the authors re-organize their results based on previously suggested grouping (age categories and risk groups)

• I suggest that the authors use sex instead of gender, unless the authors of the included studies clearly reported gender identity

Discussion

• Lines 298-305. In my opinion, it is relevant to contrast the cannabis use prevalence (based on age groups) in Iran with other countries. As previously suggested, a re-grouping of results based on relevant age-ranges could enable better comparisons with the prevalence in other countries/geographical areas.

• Lines 306-315. Discussing the prevalence of cannabis abuse/dependence/CUD is also relevant. Unfortunately, the authors have not focused on this outcome in their analyses. This could be an added value of the present review as this outcome was not included in the review published by Nazarzadeh et al. referenced above.

• Lines 316-325- Comparing the cannabis use trend with national seizures is an interesting topic. I recommend that authors provide in this paragraph the results of additional analyses conducted on this topic and not in the results section (and abstract).

• Limitations: 1) the authors should acknowledge that participants could have under-reported cannabis use as its consumption is illegal in Iran; 2) The heterogeneity was high, and results should be interpreted with caution

6. PLOS authors have the option to publish the peer review history of their article (what does this mean?). If published, this will include your full peer review and any attached files.

Reviewer #1: No

Reviewer #2: No

---

## [Author Response · Author response to Decision Letter 0]

1 Jul 2021

Journal Requirements

 Amended.

2. In the Methods section please provide additional information regarding the background and training of the experts consulted during the literature search.

Done.

3. Furthermore, please provide additional details regarding the validation of the quality assessment tool used.

We only slightly revised the well-known quality assessment tool, Joanna Briggs Institute quality assessment tool, to better suit the prevalence studies. The slightly revised version has been used in other previously published studies of our center:

Rostam-Abadi Y, Gholami J, Amin-Esmaeili M, Safarcherati A, Mojtabai R, Ghadirzadeh MR, et al. Tramadol use and public health consequences in Iran: A systematic review and meta-analysis. Addiction. 2020;115(12):2213-42.

Ansari, M., Rostam-Abadi, Y., Baheshmat, S., Hamzehzadeh, M., Gholami, J., Mojtabai, R., Rahimi-Movaghar, A. Buprenorphine abuse and health risks in Iran: A systematic review. Drug and Alcohol Dependence. 2021

4. Finally, pl¬ease provide additional details regarding how cannabis use disorder was defined as a part of the study inclusion criteria.

We added the criteria each study applied for CUD in the relative tables. Also, the eligibility criteria and data extraction sections were edited accordingly. 

 

Reviewer #1 

Methods

1) Please provide the list of reviewers in the “Eligibility criteria and screening” section (page 4).

Amended.

2) FigureS1; there were 307 excluded papers (=88%). That is the big number. Thus, the authors should provide the reasons (e.g., not quantitative empirical, no specific effect measure, not primary outcome of interest, or no full text) and also how many papers in each reason.

We applied a wide search strategy. Also, to avoid missing any relevant data, we have screened the full text of studies regarding substance use in Iran even if the relevant cannabis measures were not reported in the title/abstract. Therefore, the number of articles excluded in the stage of full-text review is high. 

We added the numbers of studies for each reason of exclusion in the Flow-diagram. We noted that 58 studies should have been excluded in the title/abstract stage and not the full-text review stage; therefore, we corrected the numbers accordingly. Due to the large number, we did not add the exclusion table, which we can provide if needed.

Results

3) Table 1: typo error “Mostly 21-0 years” – Damari, 2020.

Edited.

4) Fig S1-S11, the authors did the subgroup analysis, including by gender (male/female), frequency (lifetime, 12-mo, last month …), university student, etc. However, high heterogeneity has still been found. Testing cause of heterogeneity according to the variation of quality of included studies should be concerned in this study.

We had conducted a heterogeneity study for the total included studies (Result, Heterogeneity study section). The number of unfulfilled quality criteria showed no significant association with cannabis use prevalence (S3 Table). We added this issue (the high heterogeneity) in the limitation section. 

We had also performed a sensitivity analysis for those studies with three or more unfulfilled quality items out of the nine. There were only six with three unfulfilled items and no study with >3 unfulfilled. The results of the sensitivity analysis for relative subgroups had been presented in the Result, Quality assessment section.

5) The effect of spatiotemporal (i.e., place and time) will affect the pooled prevalence during 1990 to 2021. Table 5 can explain the temporal effect, but not for spatial effect. In my point of view, the subgroup analysis by regions should be done in this study.

The number of studies conducted among the general population was not enough, neither for temporal nor for regional subgroup analysis. 

For the studies conducted among the "combined youth groups", we had performed subgroup analysis based on the geographical regions in the Result, Geographical distribution section. As it is evident in the maps (S11 Fig), no data were available at the province level for 25 provinces. Therefore, there are not enough data for adding the regional subgroups to the current temporal analysis. We stated this shortcoming in the Limitation section.

There was an adequate number of studies only for the Tehran province (eight and nine studies in the male and female subgroups, respectively; mostly were among the university students). Therefore, we added the trend analysis for these studies (Result, Trend section, last paragraph). The method and discussion sections were edited accordingly. 

6) The authors tried to analyze the pooled prevalence during 1990 to 2021. In fact, the prevalence has been changed year by year. Thus, subgroup analysis by study year might be provide some information to the authors.

To overcome the limitation that stated in this important comment, we have sub-grouped the included studies by the study year to 5-year time intervals with details (Table 5). Furthermore, we presented all of the forest plots sorted by the study year in each subgroup. Also, in the Abstract and the Discussion, we highlighted only the latest pooled estimates for better interpretation of the current situation in Iran.  

Reviewer #2 

Dear authors,

A similar systematic review was published by Nazarzadeh et al., (2015). Prevalence of Cannabis Lifetime Use in Iranian High School and College Students: A Systematic Review, Meta-Analyses, and Meta-Regression. DOI: 10.1177/1557988314546667. In my opinion, it would be useful to comment on the added value of this review and to compare your results with the results of the mentioned review that searched for references between 1979 and 2014. 

Amended in the Discussion, first paragraph.

Please find below my suggestions to increase the accuracy of reporting. In my opinion, a re-categorization of the used groups and including cannabis use/dependence/cannabis use disorders as a separate outcome could increase the value of the manuscript.

Introduction

1) Line 33-Please clarify the statement “However, there is anecdotal evidence that cannabis use is increasing in the country and is becoming an important public health problem”.

We changed the wording of this sentence.

2) Lines 36-38- Youth is generally (e.g., by the United Nations) defined as 15-24 years and include high school and many university students. Therefore, it would be useful to revisit the three mentioned categories: youth, high-school students, and university students.

Many thanks for this important comment. We did not pre-defined the age limit before study implementation. After the final inclusion, if not reported in the full text, we requested authors of studies among the general population for the age-specific data. Finally, due to high heterogeneity in the presented age groups, we defined the age limit for the young general population subgroup in such a way not to miss any data (15-34 years). Although, as you have stated, youth is generally defined as 15-24 years, some important reports have used other categories as well. Please see: 

• Young adults as 15-34 years in "European Monitoring Centre for Drugs and Drug Addiction (2019), European Drug Report 2019: Trends and Developments, Publications Office of the European Union, Luxembourg."

• Young people as aged under 30 years in "Australian Institute of Health and Welfare 2017. National Drug Strategy Household Survey 2016: detailed findings. Drug Statistics series no. 31. Cat. no. PHE 214. Canberra: AIHW."

A notable proportion of adolescents drop out of high school in Iran. Similarly, a large group of youths does not enter university in Iran. As a result, we think these subgroups of high school students and university students are essentially different and we cannot generalize the results of high-school/university students to population groups of the same age. Therefore, in order not to miss any information and as the number of studies in these two subgroups were not low (33 studies among university students and 18 studies among high school students), we preferred to present and analyze these subgroups separately in the text, tables, and relative forest plots. 

However, for assessing the trend of the prevalence of cannabis use, we had to merge the "university students" and "high-school students" subgroups with the "young general population". To distinguish this merged group, we used the word "combined" youth group.

We added a few sentences in the Methods, Data extraction and quality assessment section, for the description of the subgroups. 

3) The authors did not provide the rationale neither for studying the prevalence of cannabis use in high risk groups nor for national seizures.

In this systematic review, we aimed not to miss any data regarding the use and use disorder of cannabis in Iran. Therefore, we included the pattern and prevalence of cannabis use among PWUD in addition to treatment-seeking for CUD among PWUD due to the important implications.

The data regarding national seizures are also provided for better interpretation of the trend of cannabis use and use disorder prevalence. As properly commented in comment No. 26, we have omitted the seizures-related statements from the Result section and Abstract and confined them to the Discussion section. 

Methods 

4) Please specify if the systematic review was registered on the International Prospective Register of Systematic Reviews (PROSPERO) or another repository.

This study was not pre-registered, and we added this in the uploaded PRISMA checklist. 

5) Lines 42- On the world scene, the landscape has significantly changed in the last 2 decades with the legalization and decriminalization of cannabis use. Therefore, it would be useful to provide a rationale for including in the search references starting 1990.

Amended.

6) I recommend that the authors provide their research question(s).

We have provided the aim of the study in more detail in the Introduction (paragraph 3).

7) a) According to the PRISMA guidelines, it is highly recommended (and necessary) to provide the study eligibility criteria in PICO format. A clear definition of the outcome appears only in the results section, b) i.e., the authors combined use prevalence with cannabis abuse/dependence/cannabis use disorder (CUD). c) In my opinion, an important (secondary) outcome would be the prevalence of cannabis abuse/dependence/CUD among cannabis users in general and among frequent users.

a) We added the research questions (according to comment No.6) in more detail in the Introduction section, paragraph 3. We did not re-state the PICO in the method section to avoid repetition.

b) We have not merged the data regarding the prevalence of use with data on cannabis use disorder. The Result section is categorized based on the target population (general population, young general population, university students, high school students, high-risk population). In each section, we separately described and presented studies on "cannabis use" and "cannabis use disorder". According to this important comment and as stated above, we have changed the paragraphing of the result section slightly and added a few sentences to increase the clarity.

c) Four studies are providing the prevalence of cannabis use disorder included in our study as the following: 

Amin-Esmaeili, 2016 Use more than 5 times in the last 12 months: 0.9% CUD: 0.5%

Ahmadi, 2003 Lifetime: 6.3% CUD: 2.5%

Yasamy, 2002 Current use: 0.7% CUD: 0.0%

Ahmadi, 2004 Lifetime: 0.2% CUD: 0.0%

As it is apparent, there is low number of cannabis users or the rate of CUD being 0, the secondary analysis of CUD among cannabis users would be misleading. Therefore, we preferred not to report this estimate.

8) The selected quality appraisal tool is adequate for observational studies of prevalence. The authors mentioned that they included studies of any methodology and design; in my opinion, it would be useful to report how the quality of intervention studies was appraised. If intervention studies were not included, this should be stated in the eligibility criteria.

Interventional studies were excluded. The exclusion criteria was completed. 

9) One of the outcomes of interest was cannabis abuse/dependence/CUD. It would be important that the authors provide additional details on how this outcome was operationalized and how many of the included studies met criterion 6 of the Joanna Briggs Institute appraisal tool “Were valid methods used for the identification of the condition?”

We included any study providing data regarding cannabis use disorder using any diagnostic criteria or definition, and we added the definition/criteria of each study in the relative tables. The unmet unfulfilled quality items are presented in all tables for each study.

10) Line 57, please include the initials of the persons involved in the screening of references and the initials of the person who mediated disagreements.

Done.

11) Lines 60-64, the authors mention that data related to prevalence use was extracted. I am unsure whether this includes cannabis abuse.

We aimed to include any data regarding any use of cannabis and cannabis use disorder. The sentence was edited.

12) Line 71- studies who reported result separate by gender were included in the meta-analyses. What happened with studies who did not report separately by sex or gender? Was this an exclusion criterion?

They were only presented in the relative tables and reported in the text and as we stated in the result section. "Studies not reporting sex-specific data were not included in the meta-analysis." We moved this sentence to the Method section, as noted in this comment for higher clarity.

13) Line 73, 80, I recommend that the authors provide a clear definition of the population subgroups; they used a mix of age and education status (students) e.g., what is the difference between young general population and general population? I suggest using groups based on relevant age-ranges (e.g., youth) as a primary outcome and high-risk groups (considering the relative low number of studies, the categories could be collapsed) as a secondary outcome.

As stated earlier, as the studies among university and high school students are essentially different in Iran, we decided to present these studies separately.

Regarding the age groups among the general population, the age group-specific data was only available for some studies. Even after requesting the authors for further data and analysis, the resultant age groups were very heterogeneous. Therefore, we created a separate group (young general population) with a wide age definition of 15-34 years, which was not pre-defined.

For trend analysis, in order to have an adequate number of studies in each time interval and for assessment of geographical distribution, we merged all studies conducted among university and high school students with the young general population to form a "combined youth group". 

We added a few sentences in the Methods for description of the subgroups. 

14) Lines 74-75 Additional details related to frequency of use are needed e.g., how is “currently the main drug” indicative of the frequency of use; what is included in last month or current?

The four national situation assessment surveys among the PWUD have reported the prevalence of "currently the main drug of use" for various substances; the cannabis data has been presented in this study (Result, High-risk groups, People who use drugs). It is mainly a measure of cannabis use disorder. We changed this section slightly for clarity.

We pooled data on last month use with current use due to the scarcity of studies reporting these measures (Limitations section), and reported this measure as "last month or current use" prevalence.

15) Line 83: Please explain the meaning of “network scale-up method”.

We added a citation to the description of network scale-up method.

16) Lines 87-89, If not enough data was available for some periods, I suggest collapsing categories e.g., 2000-2010. As no rationale was provided for selecting the 5-year time intervals, using 10-years intervals could be a viable alternative. Why was the interval 1990-2000 not used? The same observation applies to the prevalence of cannabis use.

Regarding the studies among the general population, as only two studies were reporting the prevalence of last 12-month use (in 2011 and 2015), we were not able to provide trend analysis with any time interval. 

Regarding the studies among the "combined youth groups", we have chosen the smallest possible time interval for the trend analysis with an adequate number of studies in each period. Moreover, the studies conducted before 2000 did not provide sex-specific data; therefore were not entered into the analyses. Thus, with a 10-year interval, we would not be able to analyze the trend. We highlighted this issue in the Method section. 

17) Lines 95-97, presenting data on national seizure of cannabis is interesting but it is not part of the main objectives, not sure why it was mentioned in the abstract.

We omitted seizures' related sentence from the Result section of the abstract. 

18) Line 98, please provide the name of the package used in R for meta-analyses.

Added.

Results

19.a) It is common practice to provide the PRISMA flow diagram in the main manuscript (not as an appendix). 

Done.

19.b) How can authors explain that more than half (50 out of 90) of included studies were identified by using additional resources (e.g., contacting experts). I recommend that for these additional studies, the authors report how many were initially recommended/identified and how many were excluded at each stage of reference screening (i.e., title and abstract and full text screening stages).

Unfortunately, we had not recorded the number of studies we found through additional sources. However, we provided more details on which sources we used for our opportunistic methods for higher transparency in the Methods, Search strategy section.

The underlying reason for the high number of studies included by opportunistic methods are the followings:

1) Many important and large-scale studies in the addiction field are supported by the Drug Control Headquarters of Iran. The results of these studies are not required to be published in peer-reviewed journals and are accessible as reports.

2) The Iranian database does not cover all Iranian journals. Also, there are some limitations with Boolean operators in this database, limiting the extent of our search. Therefore we overcame this limitation through contact with experts and backward citation tracking.

19.c) I recommend that the authors report the proportion of published studies out of these 50 additional references and the results of quality appraisal.

Added in the Results, first paragraph.

20) I recommend that the authors re-organize their results based on previously suggested grouping (age categories and risk groups).

Stated earlier in comments No. 2 and No. 13.

21) I suggest that the authors use sex instead of gender, unless the authors of the included studies clearly reported gender identity.

Amended.

¬Discussion

22) Lines 298-305. In my opinion, it is relevant to contrast the cannabis use prevalence (based on age groups) in Iran with other countries. As previously suggested, a re-grouping of results based on relevant age-ranges could enable better comparisons with the prevalence in other countries/geographical areas.

We have responded to this important comment on the re-grouping of the age groups in comments No.2 and No.13. We have compared our findings among the youth with the European countries, the USA, and Canada in the Discussion, paragraph 5.

23) Lines 306-315. Discussing the prevalence of cannabis abuse/dependence/CUD is also relevant. Unfortunately, the authors have not focused on this outcome in their analyses. This could be an added value of the present review as this outcome was not included in the review published by Nazarzadeh et al. referenced above.

We presented these important findings in the result section and discussed them in the Discussion, although we could not perform secondary analyses. As only three studies among the general population provided the prevalence of CUD with different study years, we confined to present the data without further analysis. Two of these studies were nationally conducted in 2001 and 2011, and "The prevalence of cannabis use disorder in national studies rose from 0% in 2001 to 0.5% in 2011." Similarly, there was only one study among the high-school students and none among the young general population and the university students on CUD. There were only two studies providing data regarding treatment-seeking for CUD, not enough for further analysis.

We compared our results with the global estimate and some examples of the US, European countries, and India in the Discussion (paragraph 3). We also added that this is the first review including studies on CUD in Iran in the Discussion (first paragraph). 

24) Lines 316-325- Comparing the cannabis use trend with national seizures is an interesting topic. I recommend that authors provide in this paragraph the results of additional analyses conducted on this topic and not in the results section (and abstract).

Amended.

25) Limitations: 1) the authors should acknowledge that participants could have under-reported cannabis use as its consumption is illegal in Iran; 2) the heterogeneity was high, and results should be interpreted with caution.

Amended. 

26) Have the authors made all data underlying the findings in their manuscript fully available?

Reviewer #2: No

All of the underlying data for this review are presented in the tables (in the main text) and all the forest plots are provided as supporting information.

---

## [Decision Letter · Decision Letter 1]

28 Jul 2021

PONE-D-21-10629R1

Evidence for an increase in cannabis use in Iran – A systematic review and trend analysis

PLOS ONE

Dear Dr. Rahimi-Movaghar,

Thank you for submitting your manuscript to PLOS ONE. After careful consideration, we feel that it has merit but does not fully meet PLOS ONE’s publication criteria as it currently stands. Therefore, we invite you to submit a revised version of the manuscript that addresses the points raised during the review process.

We look forward to receiving your revised manuscript.

Kind regards,

Chaisiri Angkurawaranon

Academic Editor

PLOS ONE

Journal Requirements:

Reviewers' comments:

Reviewer's Responses to Questions

**Comments to the Author**

1. If the authors have adequately addressed your comments raised in a previous round of review and you feel that this manuscript is now acceptable for publication, you may indicate that here to bypass the “Comments to the Author” section, enter your conflict of interest statement in the “Confidential to Editor” section, and submit your "Accept" recommendation.

Reviewer #1: All comments have been addressed

Reviewer #2: (No Response)

2. Is the manuscript technically sound, and do the data support the conclusions?

Reviewer #1: Yes

Reviewer #2: Yes

3. Has the statistical analysis been performed appropriately and rigorously? 

Reviewer #1: Yes

Reviewer #2: Yes

4. Have the authors made all data underlying the findings in their manuscript fully available?

Reviewer #1: Yes

Reviewer #2: No

5. Is the manuscript presented in an intelligible fashion and written in standard English?

Reviewer #1: Yes

Reviewer #2: Yes

6. Review Comments to the Author

Reviewer #1: (No Response)

Reviewer #2: Dear authors,

Thank you for addressing my previous comments and for providing clarifications. I appreciate your effort in synthesizing data and providing a comprehensive and up-to-date review of cannabis use in Iran. I provided additional suggestions that could increase the value of the manuscript. I encourage the authors to properly acknowledge the contribution of the work of Nazarzadeh et al., considering the paucity of reviews related to cannabis use in Iran and to pay more attention to how the methodology is reported e.g., lack of research questions, incomplete eligibility and exclusion criteria. I am well aware of the challenges associated with conducting and publishing systematic reviews and I hope that the authors will consider the suggestions provided below.

In my original report, I suggested that the authors remove from the abstract/results the analyses related to national cannabis seizures as this is not an objective of the review and keep this topic for the discussion section. The authors responded, “We omitted seizures' related sentence from the Result section of the abstract” but I am unsure why they kept following statement in the abstract “Trends of various use indicators and national seizures were examined.”

In the abstract, following statement comes out of the blue: “Treatment seeking for cannabis use disorder among those with substance use disorder attending treatment ranged from 0.9% to 10.9%”

In the introduction, the authors inaccurately state “Several studies have examined the prevalence of cannabis use along with other drugs in the general population; however, we know little about the prevalence of cannabis use in different Iranian population subgroups.” In my opinion, the authors should adequately acknowledge in the introduction and discussion sections the systematic review published by Nazarzadeh et al., (2015). Prevalence of Cannabis Lifetime Use in Iranian High School and College Students: A Systematic Review, Meta-Analyses, and Meta-Regression. DOI: 10.1177/1557988314546667. In this review, the authors provide more data related to cannabis use/dependence (by including more groups such as general population, high-risk populations) compared to the review conducted by Nazardeh et al. Consequently, I encourage the authors to elaborate on similarities and differences such as the number of studies retained (and number of participants) in the university and high-school groups, lifetime cannabis use, etc.

In my opinion, the authors should provide a clear description of the eligibility criteria. I recommend that the authors provide in a dedicated paragraph (and not in the data extraction section) a clear description of the population of interest (e.g., what “general population means” ) and of the outcomes. The authors provided in the PRISMA flow diagram exclusion criteria such as “non eligible source population” but this source population was not defined in the eligibility criteria section.

It is important that the authors clearly define what was included in the Cannabis Use Disorder (CUD) outcome. The CUD diagnostic criteria were introduced in DSM-5 (2013) and between 2000-2013 the DSM-4 used the terminology “cannabis dependence” and “Cannabis abuse”. Therefore, using CUD for studies conducted before 2013 is misleading and it would be adequate to define this outcome as cannabis dependence/abuse. In some sections (introduction, discussion) the authors refer to cannabis use disorder, in other sections (e.g., high risk groups) to cannabis dependence.

The authors provided in their response the rationale for reporting results based on different population groups i.e., differences between students and other youth. I encourage that the authors explain in the manuscript the rationale for selecting these groups. I am unsure whether the decision to analyze these groups was made before starting the systematic review (as suggested by the study aims provided in the introduction) or at the analysis phase.

The authors excluded from analyses studies that did not report cannabis use separately for men and women, but this is not stated as an exclusion criterion in the PRISMA flow diagram. Therefore, it is difficult for the reader to figure out how many studies were excluded based on this criterion.

To my understanding the population groups used are mutually exclusive e.g., general population, university students, etc. The sum of studies reported in tables 1 to 4 is 102 and does not correspond to the number of studies included in the review based on the PRISMA diagram (90 studies).

The authors acknowledged in their response that they did not keep track of the screening process for the 50 studies identified outside of the international databases search. I appreciate their efforts to identify as many eligible studies as possible, but this process lacks transparency and impedes on the reproducibility of the review. Therefore, I consider that this should be acknowledged as a limitation especially because more than half of the total number of included studies were found using this method.

As previously suggested, it would be useful for readers that the authors provide the name of the R package used for analyzing data. The authors (based on my previous suggestion) provided the name of the functions but omitted the name of the package.

As suggested in my report, it would be useful to provide a short description of the “network scale-up (NSU) method” since the authors excluded NSU studies from the analyses and to provide the number of studies that were excluded based on this criterion.

7. PLOS authors have the option to publish the peer review history of their article (what does this mean?). If published, this will include your full peer review and any attached files.

Reviewer #1: No

Reviewer #2: No

---

## [Author Response · Author response to Decision Letter 1]

4 Aug 2021

Reviewers' comments

Reviewer #1: 

All comments have been addressed.

 

Reviewer #2:

Dear authors,

Thank you for addressing my previous comments and for providing clarifications. I appreciate your effort in synthesizing data and providing a comprehensive and up-to-date review of cannabis use in Iran. I provided additional suggestions that could increase the value of the manuscript. I encourage the authors to properly acknowledge the contribution of the work of Nazarzadeh et al., considering the paucity of reviews related to cannabis use in Iran and to pay more attention to how the methodology is reported e.g., lack of research questions, incomplete eligibility and exclusion criteria. I am well aware of the challenges associated with conducting and publishing systematic reviews and I hope that the authors will consider the suggestions provided below.

Many thanks for the comprehensive review and insightful comments. Certainly, they would significantly improve this study. We have applied and responded to all the comments: 

1) In my original report, I suggested that the authors remove from the abstract/results the analyses related to national cannabis seizures as this is not an objective of the review and keep this topic for the discussion section. The authors responded, “We omitted seizures' related sentence from the Result section of the abstract” but I am unsure why they kept following statement in the abstract “Trends of various use indicators and national seizures were examined.”

We omitted this statement from the Abstract.

2) In the abstract, following statement comes out of the blue: “Treatment seeking for cannabis use disorder among those w¬ith substance use disorder attending treatment ranged from 0.9% to 10.9%”.

We omitted this sentence from the Abstract. 

3) In the introduction, the authors inaccurately state “Several studies have examined the prevalence of cannabis use along with other drugs in the general population; however, we know little about the prevalence of cannabis use in different Iranian population subgroups.” In my opinion, the authors should adequately acknowledge in the introduction and discussion sections the systematic review published by Nazarzadeh et al., (2015). Prevalence of Cannabis Lifetime Use in Iranian High School and College Students: A Systematic Review, Meta-Analyses, and Meta-Regression. DOI: 10.1177/1557988314546667. In this review, the authors provide more data related to cannabis use/dependence (by including more groups such as general population, high-risk populations) compared to the review conducted by Nazardeh et al. Consequently, I encourage the authors to elaborate on similarities and differences such as the number of studies retained (and number of participants) in the university and high-school groups, lifetime cannabis use, etc.

The relevant section in the Introduction was edited accordingly, pointing to the differences in the scope of the two studies. Moreover, the result of the previous study was added in more detail in the Discussion section. 

4) In my opinion, the authors should provide a clear description of the eligibility criteria. I recommend that the authors provide in a dedicated paragraph (and not in the data extraction section) a clear description of the population of interest (e.g., what “general population means”) and of the outcomes. The authors provided in the PRISMA flow diagram exclusion criteria such as “non-eligible source population” but this source population was not defined in the eligibility criteria section.

We moved the mentioned sentences to the "Eligibility criteria and screening" section as a separate paragraph and extended the details on the definition of the target population. The "non-eligible source population" was added to the eligibility criteria.

5) It is important that the authors clearly define what was included in the Cannabis Use Disorder (CUD) outcome. The CUD diagnostic criteria were introduced in DSM-5 (2013) and between 2000-2013 the DSM-4 used the terminology “cannabis dependence” and “Cannabis abuse”. Therefore, using CUD for studies conducted before 2013 is misleading and it would be adequate to define this outcome as cannabis dependence/abuse. In some sections (introduction, discussion) the authors refer to cannabis use disorder, in other sections (e.g., high risk groups) to cannabis dependence.

The relevant sentence in the Method section was clarified as below:

"Whatever criteria of the cannabis use disorder, either based on Diagnostic and Statistical Manual of Mental Disorders version IV or V or any other definitions, the studies were included. The applied criteria were reported precisely as stated in the study."

We stated whatever terminology the studies applied in the result section and relative tables; however, in the Introduction and Discussion sections we used "cannabis use disorder", to be consistent with the latest DSM-V. 

6) The authors provided in their response the rationale for reporting results based on different population groups i.e., differences between students and other youth. I encourage that the authors explain in the manuscript the rationale for selecting these groups. I am unsure whether the decision to analyze these groups was made before starting the systematic review (as suggested by the study aims provided in the introduction) or at the analysis phase.

Based on the previous reviews conducted by our center, studies investigating the prevalence of substance use are mainly conducted among the general population, university students, high school students, or high-risk subgroups in Iran. Therefore, we selected the target population accordingly. This rationale was added to the Method section.

7) The authors excluded from analyses studies that did not report cannabis use separately for men and women, but this is not stated as an exclusion criterion in the PRISMA flow diagram. Therefore, it is difficult for the reader to figure out how many studies were excluded based on this criterion.

As these studies were not excluded from the systematic review and were presented in the tables, we did not add this criterion to the eligibility criteria. However, we added the number of studies excluded from the meta-analysis to the Result section. 

8) To my understanding the population groups used are mutually exclusive e.g., general population, university students, etc. The sum of studies reported in tables 1 to 4 is 102 and does not correspond to the number of studies included in the review based on the PRISMA diagram (90 studies).

This is since there are "Four studies provided measures for both the general population and young general population." Also, we have presented the results of four different provinces and rounds of the Persian Youth Cohort study (six rows for one study) and the result of one repeated cross-sectional study among university students (four rows for one study) separately in the relative tables. This would add up to 102 rows for 90 studies. We added two clarifying sentences regarding the latter issue in the Result section. 

9) The authors acknowledged in their response that they did not keep track of the screening process for the 50 studies identified outside of the international databases search. I appreciate their efforts to identify as many eligible studies as possible, but this process lacks transparency and impedes on the reproducibility of the review. Therefore, I consider that this should be acknowledged as a limitation especially because more than half of the total number of included studies were found using this method.

Was added in the Limitation section. 

10) As previously suggested, it would be useful for readers that the authors provide the name of the R package used for analyzing data. The authors (based on my previous suggestion) provided the name of the functions but omitted the name of the package.

The name of the packages was added. 

11) As suggested in my report, it would be useful to provide a short description of the “network scale-up (NSU) method” since the authors excluded NSU studies from the analyses and to provide the number of studies that were excluded based on this criterion.

A brief definition was added in the Method section. The number of studies excluded accordingly was added in the Result section.

---

## [Decision Letter · Decision Letter 2]

10 Aug 2021

Evidence for an increase in cannabis use in Iran – A systematic review and trend analysis

PONE-D-21-10629R2

Dear Dr. Rahimi-Movaghar,

We’re pleased to inform you that your manuscript has been judged scientifically suitable for publication and will be formally accepted for publication once it meets all outstanding technical requirements.

Kind regards,

Chaisiri Angkurawaranon

Academic Editor

PLOS ONE

Additional Editor Comments (optional):

Reviewers' comments:

Reviewer's Responses to Questions

**Comments to the Author**

1. If the authors have adequately addressed your comments raised in a previous round of review and you feel that this manuscript is now acceptable for publication, you may indicate that here to bypass the “Comments to the Author” section, enter your conflict of interest statement in the “Confidential to Editor” section, and submit your "Accept" recommendation.

Reviewer #2: All comments have been addressed

2. Is the manuscript technically sound, and do the data support the conclusions?

Reviewer #2: Yes

3. Has the statistical analysis been performed appropriately and rigorously? 

Reviewer #2: Yes

4. Have the authors made all data underlying the findings in their manuscript fully available?

Reviewer #2: No

5. Is the manuscript presented in an intelligible fashion and written in standard English?

Reviewer #2: Yes

6. Review Comments to the Author

Reviewer #2: (No Response)

7. PLOS authors have the option to publish the peer review history of their article (what does this mean?). If published, this will include your full peer review and any attached files.

Reviewer #2: No

---

## [Editor Report · Acceptance letter]

20 Aug 2021

PONE-D-21-10629R2 

Evidence for an increase in cannabis use in Iran – A systematic review and trend analysis 

Dear Dr. Rahimi-Movaghar:

I'm pleased to inform you that your manuscript has been deemed suitable for publication in PLOS ONE. Congratulations! Your manuscript is now with our production department. 

Kind regards, 

on behalf of

Dr. Chaisiri Angkurawaranon 

Academic Editor

PLOS ONE